# `MixBridge`: Heterogeneous Image-to-Image Backdoor Attack through Mixture of Schrödinger Bridges

**Shixi Qin**[1]   **Zhiyong Yang**[1]   **Shilong Bao**[1]   **Shi Wang**[2]   **Qianqian Xu**[2]   **Qingming Huang**[1 3 2]

## Abstract

This paper focuses on implanting multiple heterogeneous backdoor triggers in bridge-based diffusion models designed for complex and arbitrary input distributions. Existing backdoor formulations mainly address single-attack scenarios and are limited to Gaussian noise input models. To fill this gap, we propose `MixBridge`, a novel diffusion Schrödinger bridge (DSB) framework to cater to arbitrary input distributions (taking I2I tasks as special cases). Beyond this trait, we demonstrate that backdoor triggers can be injected into `MixBridge` by directly training with poisoned image pairs. This eliminates the need for the cumbersome modifications to stochastic differential equations required in previous studies, providing a flexible tool to study backdoor behavior for bridge models. However, a key question arises: can a single DSB model train multiple backdoor triggers? Unfortunately, our theory shows that when attempting this, the model ends up following the geometric mean of benign and backdoored distributions, leading to performance conflict across backdoor tasks. To overcome this, we propose a Divide-and-Merge strategy to mix different bridges, where models are independently pre-trained for each specific objective (`Divide`) and then integrated into a unified model (`Merge`). In addition, a Weight Reallocation Scheme (WRS) is also designed to enhance the stealthiness of `MixBridge`. Empirical studies across diverse generation tasks speak to

the efficacy of `MixBridge`. The code is available at: `https://github.com/qsx830/MixBridge`.

**Warning: This paper contains model outputs that may be offensive in nature.**

## 1. Introduction

Diffusion models have demonstrated remarkable performance across various domains, including image (Ho et al., 2020; Dhariwal & Nichol, 2021; Ho & Salimans), text (Lou et al.), audio (Kong et al.), speech generation (Huang et al., 2022), and others (Wen et al., 2022). Alongside the success of diffusion models, potential risks have emerged, such as vulnerability to backdoor attacks. In a backdoor attack scenario, the attacker secretly implants triggers into a victim model. Once unsuspecting users deploy the victim model in their applications, these triggers can be activated to produce harmful content (e.g., Not-Suitable-For-Work images).

So far, various backdoor attack methods have been developed to reveal the mechanism of backdoor attacks in diffusion models. Existing research primarily focuses on two specific formulations: unconditional diffusion models (Chou et al., 2023; Chen et al., 2023a; Chou et al., 2024) and conditional Text-to-Image (T2I) diffusion models (Zhai et al., 2023; Struppek et al., 2023; Shan et al., 2024; Wang et al., 2025). However, we argue that two crucial scenarios remain unexplored in the current literature. First, existing backdoor formulations are designed for diffusion-based models that exclusively take **Gaussian** noise as input. Unfortunately, real-world tasks often require models to handle more **arbitrary** and complex input distributions, rather than pure noise. For example, tasks like super-resolution and image inpainting use images as input in an Image-to-Image (I2I) generation context. Second, most studies merely focus on the **single** backdoor attack. Yet in practice, malicious attackers may consider **heterogeneous** backdoor attacks, such as injecting multiple backdoors with different triggers simultaneously, to enhance stealthiness and maximize the impact of their attack.

To address these issues, our goal in this paper is two-fold:

---

[1]School of Computer Science and Technology, University of Chinese Academy of Sciences, Beijing 101408, China [2]Key Laboratory of Intelligent Information Processing, Institute of Computing Technology, Chinese Academy of Sciences, Beijing 100190, China [3]Key Laboratory of Big Data Mining and Knowledge Management (BDKM), University of Chinese Academy of Sciences, Beijing 101408, China. Correspondence to: Zhiyong Yang <yangzhiyong21@ucas.ac.cn>, Qingming Huang <qmhuang@ucas.ac.cn>.

*Proceedings of the 42$^{nd}$ International Conference on Machine Learning*, Vancouver, Canada. PMLR 267, 2025. Copyright 2025 by the author(s).

> **1)** *Investigate I2I backdoor attacks on diffusion models across arbitrary input and output image distributions.* **2)** *Explore heterogeneous backdoor attacks in this context.*

For **1)**, our proposed method is on top of the bridge-based diffusion models (De Bortoli et al., 2021; Zhou et al.; Gushchin et al., 2024; Yang et al., 2021), which have garnered significant attention due to their flexibility in achieving transformations between arbitrary distributions. Specifically, we propose MixBridge, an MoE-based I2I Schrödinger bridge (I2SB) model (De Bortoli et al., 2021; Liu et al., 2023; Shi et al., 2024). Different from traditional diffusion models, our proposed method can now handle arbitrary input image distributions. This completely avoids the need to implant triggers on the noises. Moreover, we theoretically show that one only needs to train the MixBridge model directly with backdoor images (Prop. 4.1), where the stochastic differentiable equations (SDEs) could be implicitly learned. This differs significantly from previous studies (Chou et al., 2023; Chen et al., 2023a; Chou et al., 2024; Bao et al., 2025), which all require the cumbersome design of SDEs to implement diffusion backdoor attacks.

**For 2)**, we find that it is challenging to execute heterogeneous backdoor attacks within a single I2SB model. Specifically, if one uses a single model to fit the sample path of the benign distribution and all different types of backdoored distribution, then we prove that (Thm. 4.2) the algorithm follows the **geometric mean** of these distributions. Because the sample paths for benign and different backdoor tasks differ greatly in distribution, the geometric mean and most target distributions often **lies from afar**, degrading generation quality for both benign and malicious generation tasks. To resolve this challenge, we propose a **divide-and-merge** strategy to reconcile the performance conflict between different tasks. In the **divide** stage, we use a task-specific warm-up strategy to pre-train different tasks separately. In the **merge** stage, we integrate the pre-trained model with the Mixture of Experts (MoE) framework. Another key issue is that a naive MoE framework tends to assign a much higher weight to the original pre-trained expert for the given tasks. By looking at the weight assignment statistics, the user can easily detect which experts are responsible for the backdoor attacks. To further improve the stealthiness of MixBridge, we propose a Weight Reallocation Scheme (WRS) to encourage uniform weight assignment, making the backdoor experts less noticeable to the victim user. In this sense, MixBridge can retain the advantages of task-specific models while keeping the weight assignments inconspicuous.

Finally, we validate the effectiveness of MixBridge on the ImageNet and CelebA datasets. Our results demonstrate the model's dual capabilities: producing high-quality be-

nign outputs when given clean input images (i.e., utility) and generating heterogeneous malicious outputs when input images contain triggers (i.e., specificity).

Our contributions can be summarized in three folds:

- We propose MixBridge, an I2I diffusion Schrödinger Bridge (DSB) model, to study the backdoor attacks. To our knowledge, we are the first to study the backdoor attacks on the arbitrary diffusion bridge models.

- We propose a divide-and-merge training strategy to reconcile the performance conflict between benign and different malicious generation tasks.

- We validate the performance of the proposed framework on ImageNet and CelebA, demonstrating that our model can generate high-quality benign and heterogeneous malicious outputs.

## 2. Related Work

**Diffusion Model.** The diffusion model is widely used in image generation (Song et al., a;b; Meng et al., 2024). While early diffusion models (Ho et al., 2020) were designed for unconditional image generation, subsequent models incorporated additional guidance to enhance control over the generated content (Dhariwal & Nichol, 2021; Chao et al., 2021). Based on these, researchers propose Image-to-Image (I2I) diffusion models, which take input images and generate guided outputs (Nichol et al., 2022; Sasaki et al., 2021). (Saharia et al., 2022a) applies conditional diffusion models to edit images, and (Saharia et al., 2022b) uses a stochastic iterative denoising process to achieve super-resolution. Apart from these, some studies have explored diffusion bridges for image generation (Shi et al., 2024; Zhou et al.). For instance, the diffusion Schrödinger Bridge (DSB) achieves an optimal transport between two distributions, suitable for I2I generation. In (Ye et al., 2024), DSB is applied to solve the unpaired I2I translation between distinct image distributions. (Wang et al., 2024b) employs DSB to perform super-resolution for medical images.

**Backdoor Attacks for Diffusion Models.** Early studies on backdoor attacks mainly focus on the classification tasks (Gu et al., 2017; Doan et al., 2021), where a backdoored model produces predefined predictions only when the input contains the trigger. Recently, researchers have investigated backdoor attacks on different generative models (Rawat et al., 2022; Jiang et al., 2024). In this paper, we restrict our discussion to the diffusion-based models, which aim to modify diffusion models so that the attacker can manipulate outputs by activating hidden backdoors. Regarding the diffusion model, previous studies can be roughly divided into Text-to-Image (T2I) and unconditional diffusion attacks.

The T2I diffusion model involves additional text as input, so the attacker can easily inject triggers into the text (Zhai et al., 2023; Struppek et al., 2023; Wang et al., 2025; Pan et al., 2023; Huang et al., 2023). Previous backdoor attack methods on unconditional diffusion models (Chou et al., 2023; 2024; Chen et al., 2023a) generate benign images from a standard Gaussian noise and inject triggers into the Gaussian noise to induce the model to generate backdoor outputs. In response to backdoor attacks, some defense algorithms have been proposed (An et al., 2024; Guan et al., 2024; Mo et al., 2024; Yang et al., 2023b; 2022), which detect backdoor attacks by analyzing the input noise distributions.

**A Summary of Ours.** The relationship between our proposed backdoor attack and prior studies can be concluded as follows. In terms of the similarities, our backdoor attack method is built upon the generative diffusion framework, which solves I2I tasks in the benign case, and generates malicious outputs if the input contains a trigger. However, it differs in four key aspects. First, unlike early backdoor attacks that induce misclassifications, the MixBridge aims to generate target backdoor images. Second, the MixBridge targets the bridge-based diffusion models that directly take images as inputs, while previous studies mainly focus on unconditional or T2I diffusion models that start from a standard Gaussian noise. Third, we consider heterogeneous backdoor attacks against diffusion models, while previous studies consider a single backdoor attack only. Fourth, existing defenses for diffusion models rely on the assumption that the diffusion process begins with Gaussian noise. As a result, they may not effectively mitigate our proposed I2I backdoor attack.

## 3. Preliminary

In this paper, built upon the bridge-based diffusion model, we aim to investigate the I2I backdoor attacks as a representative case of the general diffusion process between arbitrary distributions. In this section, we will first introduce the I2I Schrödinger bridge we implement and then outline the setups for the backdoor attacks.

### 3.1. Image-to-Image Schrödinger bridge

Our work is primarily built on a recently emerging effective diffusion bridge model called the Image-to-image Schrödinger bridge (I2SB) (Liu et al., 2023). The goal of I2SB is to learn the nonlinear diffusion bridge between two given distributions, which can be used in various downstream image restoration tasks, such as super-resolution and image inpainting. To be specific, let $x_1 \sim p_B$ represent the input image of I2SB and $x_0 \sim p_A$ represent the corresponding target image[1]. The I2SB model $\epsilon(x_t, t; \theta)$

---

[1]The generation process is reversed from $t = 1$ to $0$.

parameterized by $\theta$ can be trained as follows:

$$\mathcal{L}(\theta) = \mathbb{E}_{t \sim \mathcal{U}(0,1), x_t, x_0, x_1} \left( \left\| \epsilon(x_t, t; \theta) - \frac{x_t - x_0}{\sigma_t} \right\|^2 \right),$$
(1)

where $x_t \sim q(x_t|x_0, x_1)$ can be considered as a linear combination of $x_0$ and $x_1$. After training, the target examples can be sampled in the same way as the standard diffusion model according to DDPM (Ho et al., 2020). This way, the target image $x_0$ can be recursively deduced from the input image $x_1$. **Details** of the DSB problem and I2SB **are discussed in Appendix** A.

### 3.2. Heterogeneous Backdoor Attacks

The backdoor attacker's goal is to manipulate a diffusion model's output to produce malicious results. Previous studies have explored single-pattern backdoor attacks in diffusion models and some associated defense methods. However, in real scenarios, a model is likely to face attacks in various patterns when an aggressive attacker injects multiple, distinct triggers into the model, which we call **heterogeneous attacks**. Seeing this new threat, we explore heterogeneous backdoor attacks on top of diffusion bridge models to deepen our understanding of the inner mechanism. To begin, we outline the attacker's setup as follows.

**Attacker's Capacity and Goal:** We assume that the attacker has full control over the training dataset and the training process. Once the model is trained, the attacker then uploads the poisoned model to a public marketplace. When people deploy this poisoned model, the attacker aims to activate the embedded backdoors to generate malicious images predetermined by the attacker while keeping others unaware. These malicious outputs may cause ethical and legal issues for the user hosting the model.

To achieve the goal, the model must satisfy two critical requirements as discussed in previous studies: **utility** and **specificity** (Chou et al., 2023; 2024). The utility requires the poisoned model to behave normally when processing clean inputs, generating high-quality benign outputs. This ensures the poisoned model does not arouse suspicion. On the other hand, when the input images contain triggers, the specificity enables the model to produce malicious outputs, such as Not-Suitable-For-Work (NSFW) images (Zhang et al., 2025), copyright infringement images (Wang et al., 2024a), etc.

## 4. **MixBridge**

In this section, we first introduce the backdoor attack problem for diffusion models. Then, we study how to inject backdoor triggers into the typical diffusion bridge model (I2SB). Taking a step further, we explore how to conduct het-

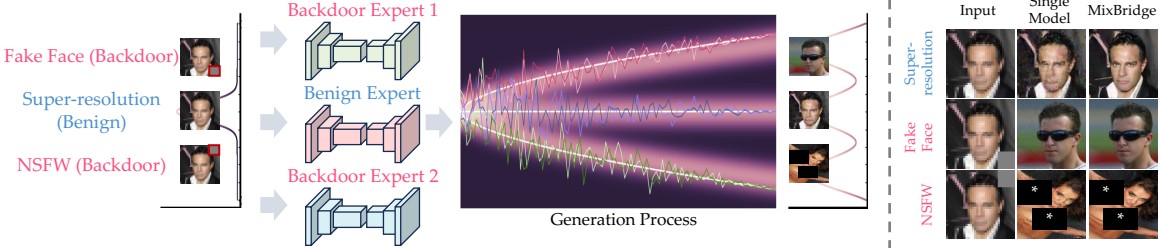

Figure 1: **Overview of the generation process of the diffusion model.** (*Left*) The model processes images from an input distribution and generates output images along distinct diffusion trajectories. Notably, both the input and output distributions are mixture distributions. While the input images maintain a degree of similarity, there exists a significant disparity among the heterogeneous output distributions. (*Right*) With various $\boldsymbol{\delta}_i$ injected into the input image, the model generates different outputs. Obviously, the quality of the output image generated by the `MixBridge` is much better than the single model.

erogeneous backdoor attacks in this context. Due to space constraints, all proofs are deferred to Appendix B.

### 4.1. Problem Formulation

In a typical diffusion process over $t \in [0, 1]$, the model processes a **c**lean input image $\boldsymbol{x}_1^c$ to generate a benign output image $\boldsymbol{x}_0^c$. To manipulate the model, an attacker may inject a backdoor trigger $\boldsymbol{\delta}$ into $\boldsymbol{x}_1^c$, creating a **p**oisoned input $\boldsymbol{x}_1^p$. This input then guides the model to produce a malicious output image $\boldsymbol{x}_0^p$ as specified by the attacker. Mathematically, this poisoning process is formalized as follows:

$$\boldsymbol{x}_1^p = (1 - \boldsymbol{M}) \odot \boldsymbol{x}_1^c + \boldsymbol{M} \odot \boldsymbol{\delta}, \qquad (2)$$

where $\boldsymbol{M}$ denotes the binary mask corresponding to the trigger, and $\boldsymbol{x}_1^p$ is the resulting poisoned input. Typically, backdoor triggers are subtle and visually inconspicuous, such as a small black patch in the corner of the input image, as illustrated in Fig. 1 (*left*).

The discussion above assumes homogeneous attacks, where a single type of backdoor is deployed. However, real-world scenarios often involve heterogeneous attacks, where multiple backdoor triggers coexist within the same poisoned model. In this setting, the model must simultaneously learn the clean generative process $\boldsymbol{x}_1^c \rightarrow \boldsymbol{x}_0^c$ and $M$ distinct backdoor processes. Formally, the $i$-th backdoor process is represented as $\boldsymbol{x}_1^{p,i} \rightarrow \boldsymbol{x}_0^{p,i}$, where the poisoned input $\boldsymbol{x}_1^{p,i}$ is obtained by blending the clean input $\boldsymbol{x}_1^c$ with a trigger $\boldsymbol{\delta}_i$ similar to Eq. 2. In our method, different triggers are strategically applied to distinct locations (i.e., using different binary masks $\boldsymbol{M}_i$) to differentiate between attacks.

As described in Eq. 1, we prepare pairs of images to train the DSB model. We use $D_c = \{(\boldsymbol{x}_{0,j}^c, \boldsymbol{x}_{1,j}^c)\}_{j=1}^{N_c}$ and $D_i = \{(\boldsymbol{x}_{0,j}^{p,i}, \boldsymbol{x}_{1,j}^{p,i})\}_{j=1}^{N_i}$ to denote the clean and $i$-th poisoned training datasets, containing $N_c$ and $N_i$ image pairs respectively. The image pairs for benign generation can be easily obtained. For example, the image pairs for the super-

resolution task are naturally the high-resolution images and the corresponding low-resolution images.

However, there is no innate relationship between clean and malicious images. For each backdoor attack, we prepare a malicious dataset $D_m = \{\boldsymbol{m}_j^{p,i}\}_{j=1}^{N_m}$. To form appropriate pairs for attacks, we select the malicious image $\boldsymbol{x}_{0,j}^{p,i}$ from $D_m$ for each clean input based on the proximity evaluated by Euclidean distance: $\boldsymbol{x}_{0,j}^{p,i} = \arg \min_{k \in D_m} \|\boldsymbol{x}_{1,j}^{p,i} - k\|_2$.

Given the settings above, we argue that a critical challenge in this context is managing the mixed data distribution of clean and poisoned images. The overall data distribution at time step $t$ is modeled as:

$$p(\boldsymbol{x}_t) = p(c) \cdot p(\boldsymbol{x}_t^c | c) + \sum_{i=1}^{M} p(i) \cdot p\left(\boldsymbol{x}_t^{p,i} | i\right), \quad (3)$$

where $p(\boldsymbol{x}_t)$ represents the probability density function of the mixture distribution at time $t$. Here, $\boldsymbol{x}_t^c$ and $\boldsymbol{x}_t^{p,i}$ denote the clean and poisoned images under the $i$-th attack, respectively. The terms $p(c)$ and $p(i)$ are proportion for the clean and $i$-th poisoned images, with the constraint $p(c) + \sum_{i=1}^{M} p(i) = 1$ ensuring a valid distribution.

How to deal with such a mixed sample path? Next, we will introduce our method, starting with a naive solution, and then propose the final heterogeneous method.

### 4.2. A Naive Method for Heterogenous Backdoor Injection

Existing methods often require extensive modifications to both the forward and reverse diffusion processes in denoising diffusion models to execute backdoor attacks, enabling the model to learn an undesired correlation between the backdoor trigger and the backdoor target. Fortunately, on top of the diffusion bridge model I2SB, we theoretically reveal that the backdoor attacks towards bridged-based models can be implemented more easily:

**Proposition 4.1** (Image generation with pair relationship). *Given the image pairs $(\boldsymbol{x}_0^{p,i}, \boldsymbol{x}_1^{p,i})$ or $(\boldsymbol{x}_0^c, \boldsymbol{x}_1^c)$ in the training datasets, the ground-truth sample-path of I2SB always generate images consistent with pairwise relationships with $t \rightarrow 1$ and $t \rightarrow 0$.*

Prop.4.1 demonstrates that the sample path of I2SB automatically connects the input and output distribution for I2I generation. In this sense, we no longer need to design specific SDEs to cater to different backdoor generation tasks. In other words, the backdoor triggers can be embedded by simply training the model with poisoned image pairs. As a **naive** solution, one only needs to **simply poison or add a subset of the training data with malicious triggers**. Then, given a clean dataset $D_c$ and various types of poisoned image pairs $D_i, i \in [M]$, we can employ the following objective function to train a backdoored I2SB model:

$$\mathcal{L}_{\text{backdoor}}(\boldsymbol{\theta}) = \textcolor{blue}{\ell^c} + \sum_{i=1}^{M} \textcolor{magenta}{\ell^i}, \tag{4}$$

where

$$\textcolor{blue}{\ell^c} = p(c) \cdot \mathbb{E}_{t, \boldsymbol{x}^c, \boldsymbol{x}_0^c, \boldsymbol{x}_1^c | c} \left\| \epsilon(\boldsymbol{x}_t^c, t; \boldsymbol{\theta}) - \frac{\boldsymbol{x}_t^c - \boldsymbol{x}_0^c}{\sigma_t} \right\|^2,$$

$$\textcolor{magenta}{\ell^i} = p(i) \cdot \mathbb{E}_{t, \boldsymbol{x}_t^{p,i}, \boldsymbol{x}_0^{p,i}, \boldsymbol{x}_1^{p,i} | i} \left\| \epsilon(\boldsymbol{x}_t^{p,i}, t; \boldsymbol{\theta}) - \frac{\boldsymbol{x}_t^{p,i} - \boldsymbol{x}_0^{p,i}}{\sigma_t} \right\|^2.$$

The key issue with this method is that it neglects the model fitting ability. Although I2SB guarantees a flexible sample path, we also need to consider **whether the expressivity of the employed model is sufficient to fit such a sample path**. Unfortunately, for a complicated heterogeneous backdoor attack, we find that the sample paths can hardly be fitted by a **single** I2SB model. In the setting of backdoor attacks, the model is required to simultaneously generate a clean output $\boldsymbol{x}_0^c$ and a poisoned malicious output $\boldsymbol{x}_0^{p,i}$. The challenge arises because $\boldsymbol{x}_0^c$ and $\boldsymbol{x}_0^{p,i}$ have significantly different distributions, whereas their respective inputs $\boldsymbol{x}_1^c$ and $\boldsymbol{x}_1^{p,i}$ differ only by a small mask $\boldsymbol{\delta}_i$, leading to mutual interference between different tasks.

Below, we present a theoretical analysis of its limitations:

**Theorem 4.2** (Limitations of using a single I2SB model for heterogeneous backdoor attacks). *Given an arbitrary image $\boldsymbol{x}_0$, the posterior of a trained I2SB model can be formulated as $\tilde{p}_{\boldsymbol{\theta}}(\boldsymbol{x}_t|\boldsymbol{x}_0)$. If we assume $\nabla_{\boldsymbol{x}_t} \epsilon_{\boldsymbol{\theta}}(\boldsymbol{x}_t, t; \boldsymbol{\theta})$ possesses full column rank[2], the posterior is proportional to the Geometric Average of the mixture distribution of all generation tasks, i.e.:*

$$\tilde{p}_{\boldsymbol{\theta}}(\boldsymbol{x}_t|\boldsymbol{x}_0) \propto \prod_i p(\boldsymbol{x}_t|\boldsymbol{x}_0, i)^{p(i|\boldsymbol{z})}, \tag{5}$$

---

[2]It is plausible that $\nabla_{\boldsymbol{x}_t} \epsilon_{\boldsymbol{\theta}}(\boldsymbol{x}_t, t; \boldsymbol{\theta})$ has full column rank, given that the number of parameters in the model significantly exceeds the dimensionality of the image feature space.

*where $p(i|z) = p(i|\boldsymbol{x}_t, \boldsymbol{x}_0, \boldsymbol{x}_1^i)$, where $i$ refers to a specific member among the clean and backdoored distributions.*

Thm.4.2 demonstrates that when a single I2SB model is used to fit the sample paths of both the benign distribution $\boldsymbol{x}_1^c \rightarrow \boldsymbol{x}_0^c$ and all heterogeneous backdoor distributions $\boldsymbol{x}_1^{p,i} \rightarrow \boldsymbol{x}_0^{p,i}$, the model tends to **approximate the geometric average of these distributions**. Given the significant disparity between the clean and backdoored distributions, this geometric mean often deviates greatly from most target distributions, leading to performance degradations for both benign and malicious tasks. We conduct a simple experiment illustrated in Fig. 1 (*right*) to highlight the limitation of this, where a single model struggles to generate satisfactory outputs for both clean and poisoned images.

What if we integrate the generation ability from multiple models? In the upcoming discussions, we will present a solution in this manner.

### 4.3. Exploring Heterogeneous Backdoor Attacks Beyond a Single Model

To achieve the performance balance across different tasks, we propose a divide-and-merge strategy borrowing the idea from the Mixture of Experts (MoE) mechanism (Chen et al., 2023b; Ma et al., 2018; Du et al., 2024). The critical point is we first train well-performing I2SB models tailored to individual tasks independently and subsequently integrate them into a unified MixBridge model. The overall framework of the proposed MixBridge is illustrated in Fig. 1 (*left*).

**Stage 1: Divide.** We propose a task-specific warm-up strategy to pre-train different tasks independently, where each task corresponds to an I2SB model. Specifically, let the clean I2SB model be denoted as $\epsilon_c(\cdot, \cdot; \boldsymbol{\theta}_c)$ and the backdoored I2SB experts be denoted as $\epsilon_i(\cdot, \cdot; \boldsymbol{\theta}_i), i \in [M]$, totally $M+1$ experts. In this stage, each model $\epsilon_*$ is trained independently with its corresponding paired images to minimize Eq. 1. For the benign I2I generation task, the expert $\epsilon_c$ is trained exclusively on clean paired images $D_c$. Similarly, the backdoored expert $\epsilon_i, i \in [M]$ is trained on the $i$-th poisoned training dataset $D_i$. This stage ensures that each model excels in its specific generation task without interference from others, while also enhancing the overall diversity of the models.

**Stage 2: Merge.** After the dividing stage, we merge all pre-trained experts as a mixture model. This requires encouraging cooperation among these experts and preserving their individual generative capabilities as much as possible. To do this, we introduce a **learnable** expert router r to adaptively determine the contribution of each model to the final output, represented by $\boldsymbol{w} = (w_c, w_1, ..., w_M)^\top \in \mathbb{R}^{(M+1)}$. Considering that backdoor attacks are typically triggered by input images, we design the router to distinguish high-level

Table 1: **Experimental results for super-resolution and per-task backdoor attacks on the CelebA dataset**. The backdoor attack results are reported as the average across all individual backdoor tasks. Note that the **purple** cell represents the optimal value, and the **green** cell represents the sub-optimal value. **SR.**: Super-resolution. **Entro. (Steal.)**: Entropy (Stealthiness).

| | SR. (Benign) | Fake Face | NSFW | Anime NSFW | Super-resolution Evaluation (Benign) | | | Per-Task Backdoor Average | | | |
|---|---|---|---|---|---|---|---|---|---|---|---|
| | | | | | FID↓ | PSNR↑ | SSIM (E-02)↑ | MSE (E-02)↓ | CLIP (E-02)↑ | ASR↑ | Entro. (Steal.)↑ |
| I2SB | ✓ | | | | 72.59 | **27.55** | 81.72 | 36.71 | 58.42 | 0.00 | 0.00 |
| Single Model | ✓ | ✓ | | | 132.83 | 25.64 | 71.13 | 27.16 | 66.63 | 32.77 | 0.00 |
| | ✓ | | ✓ | | 135.82 | 25.71 | 71.62 | 25.25 | 67.44 | 32.50 | 0.00 |
| | ✓ | | | ✓ | 134.46 | 25.21 | 67.37 | 22.43 | 65.32 | 30.10 | 0.00 |
| | ✓ | ✓ | ✓ | | 143.42 | 25.38 | 69.30 | 16.00 | 73.23 | 62.98 | 0.00 |
| | ✓ | ✓ | | ✓ | 158.23 | 25.18 | 68.19 | 12.68 | 69.47 | 53.69 | 0.00 |
| | ✓ | | ✓ | ✓ | 159.85 | 25.33 | 68.99 | 11.60 | 71.54 | 52.11 | 0.00 |
| | ✓ | ✓ | ✓ | ✓ | 161.35 | 24.98 | 66.19 | 3.05 | 71.59 | 60.91 | 0.00 |
| M.B. (Ours) — w/o WRS | ✓ | ✓ | | | 41.48 | 27.46 | **83.35** | 27.06 | 69.77 | 33.17 | 4e-03 |
| | ✓ | | ✓ | | **41.20** | 27.33 | 81.54 | 25.10 | 71.16 | 33.19 | 3e-03 |
| | ✓ | | | ✓ | 42.26 | 27.29 | 81.40 | 22.11 | 70.80 | 33.04 | 8e-03 |
| | ✓ | ✓ | ✓ | | 61.73 | 26.72 | 83.18 | 13.13 | 83.03 | 63.79 | 3e-03 |
| | ✓ | ✓ | | ✓ | 61.25 | 25.74 | 76.00 | 11.06 | 74.68 | 62.63 | 2e-03 |
| | ✓ | | ✓ | ✓ | 63.51 | 26.63 | 83.00 | 11.74 | 78.32 | 60.65 | 1e-03 |
| | ✓ | ✓ | ✓ | ✓ | 71.84 | 25.68 | 82.33 | 1.17 | 88.85 | 96.45 | 7e-03 |
| w WRS | ✓ | ✓ | | | 60.65 | 26.43 | 82.17 | 27.04 | 69.60 | 33.25 | 0.99 |
| | ✓ | | ✓ | | 68.12 | 26.50 | 80.57 | 25.24 | 71.05 | 33.08 | 0.99 |
| | ✓ | | | ✓ | 66.59 | 26.70 | 80.59 | 22.08 | 70.57 | 33.19 | 0.99 |
| | ✓ | ✓ | ✓ | | 80.88 | 24.61 | 73.37 | 15.83 | 79.46 | 66.04 | 1.57 |
| | ✓ | ✓ | | ✓ | 83.85 | 24.07 | 71.07 | 12.48 | 78.85 | 65.75 | 1.57 |
| | ✓ | | ✓ | ✓ | 77.21 | 24.99 | 73.69 | 10.88 | 81.74 | 65.84 | 1.57 |
| | ✓ | ✓ | ✓ | ✓ | 85.88 | 24.36 | 70.40 | **1.13** | **88.94** | **96.98** | **1.99** |

backdoor patterns embedded in the input features $x_1^*$. For generality, $x_1^*$ represents either clean images or any type of poisoned image. Concretely, the router r calculates the normalized weights as follows:

$$w = \mathrm{r}(x_1^*) = \mathrm{Softmax}(W^\top \mathrm{F}(x_1^*) + b), \qquad (6)$$

where $\mathrm{F}(\cdot) \in \mathbb{R}^d$ is a deep neural network (e.g., ResNet (He et al., 2016)) used to extract useful features, $d$ is the feature dimension; $W \in \mathbb{R}^{d \times (M+1)}$ is the learnable transformation matrix, and $b \in \mathbb{R}^{M+1}$ is the bias term.

Subsequently, the output of the MixBridge at each step $t$ ($t \in [0,1]$) is computed as the weighted sum:

$$\epsilon_{\mathrm{Mix}}(x_t^*, t; \theta_{\mathrm{Mix}}) = w_c \cdot \epsilon_c(x_t^*, t; \theta_c) + \sum_{i=1}^{M} w_i \cdot \epsilon_i(x_t^*, t; \theta_i), \qquad (7)$$

where $x_t^*$ is the generated image at time $t$, and again $\epsilon_c(x_t^*, t; \theta_c)$ and $\epsilon_i(x_t^*, t; \theta_i)$ are the denoising predictions of the clean and backdoored experts, respectively.

Equipped with Eq.6 and Eq.7, we proceed to fine-tune MixBridge using all datasets $D_c$ and $D_i$ ($i \in [M]$) to reconcile performance conflicts between tasks. Ideally, given different inputs (clean or backdoored), MixBridge now can adaptively approximate the optimal diffusion trajectory by assigning a higher weight to the relevant model.

**Weight Reallocation Scheme.** However, in practice, we observe that **simply merging these multiple experts is insufficient**. To minimize the reconstruction error (Eq. 1), the router r tends to assign a higher weight to the model specifically pre-trained for a particular $k$-th generation task during the divide stage, effectively setting $w_k = 1$ while assigning $w_j = 0$ for all $j \neq k$ (Refer to Sec. 5.4). This behavior undermines the model's stealthiness, as users can detect anomalies by inspecting the components of MixBridge.

To address this issue, we propose a Weight Reallocation Scheme (WRS) to prevent the weight assignment from being too far from uniform:

$$\mathcal{L}_{WRS} = \mathbb{E}_w \left[ \left\| w - \frac{1}{M+1} \right\|^2 \right]. \qquad (8)$$

Intuitively, Eq. 8 encourages the router r to assign uniformly smooth contributions to each expert, enhancing the stealthiness of the MixBridge model.

Finally, we optimize the following objective function during the merging stage to balance generative performance and backdoor stealthiness:

$$\mathcal{L}(\theta_{\mathrm{Mix}}) = \mathcal{L}_{\mathrm{backdoor}}(\theta_{\mathrm{Mix}}) + \lambda \mathcal{L}_{WRS} \qquad (9)$$

where $\boldsymbol{\theta}_{\texttt{Mix}} = (\boldsymbol{\theta}_c, \boldsymbol{\theta}_1, ..., \boldsymbol{\theta}_M, \boldsymbol{\theta}_r)$ represents all trainable parameters included in `MixBridge`, $\boldsymbol{\theta}_r$ is the learnable parameters of the router r; $\lambda$ is a tunable trade-off hyperparameter. Detailed training and generation procedures are provided in Alg. 1 and Alg. 2, respectively.

# 5. Experiments

## 5.1. Experimental Setup

**Datasets and Models.** We evaluate our proposed backdoor attack method in two normal tasks, **super-resolution** and **image inpainting**.

The experiments of super-resolution are conducted on the CelebA dataset (Liu et al., 2015). The images are resized to $128 \times 128$, and then the images are further corrupted to $32 \times 32$ by a pool filter to create low-resolution input images. The model is trained to recover the high-resolution image ($128 \times 128$) from the low-resolution input ($32 \times 32$). The experiments of image inpainting are conducted on the ImageNet $256 \times 256$ (Deng et al., 2009). We use the *20%-30%* freeform masks from (Saharia et al., 2022a) to corrupt the images for the inpainting task.

We prepare $M = 3$ backdoor attacks: Fake Face, NSFW, and Anime NSFW. For each backdoor attack, we prepare $N_m = 10$ malicious images for $D_m$ to ensure the diversity of the backdoor attacks. The Fake Face alters an input image to generate a completely different face, while the other two attacks generate pornographic outputs. We manually select ten images of the same person from the CelebA dataset for the Fake Face attack. For the latter two, we randomly select ten NSFW images and ten Anime NSFW images from (Yang et al., 2023a)[3] and Hugging Face[4].

For a fair comparison, we adopt the same model architecture and parameter settings as in (Liu et al., 2023). Each expert $\epsilon_*$ in the `MixBridge` is a UNet block pre-trained on ImageNet $256 \times 256$ (Dhariwal & Nichol, 2021). The router r is implemented as a ResNet50 model.

## 5.2. Evaluation Metrics

We evaluate our backdoor attack method from two perspectives, **utility** and **specificity**.

**Utility:** We use **FID** (Heusel et al., 2017), **PSNR**, and **SSIM** (Wang et al., 2004) to evaluate the utility of super-resolution and image inpainting tasks. A low **FID** indicates that the output image closely matches the input data distribution. The high **PSNR** and **SSIM** values indicate the

recovered image preserves the original image structure.

**Specificity:** We use **MSE**, **CLIP score** (Hessel et al., 2021), *Attack Success Rate* (**ASR**) (the ratio of successful attacks to total attacks) and Shannon information **Entropy** (Diaz et al., 2002; Murdoch, 2013) for backdoor attack evaluation. A low **MSE** implies the output image is pixel-wise close to the predefined malicious image. For the **CLIP score**, we apply a CLIP model to extract the image embeddings and compute the cosine similarity. A high CLIP score signifies the generated image's features resemble those of the malicious image. For the **ASR**, we consider a backdoor attack successful if the CLIP score exceeds a specific threshold. A high ASR indicates the model is sensitive to triggers, leading to successful attacks. In addition, we assess the stealthiness of backdoor attacks using **Entropy** (Entro.), computed with the weight distribution: $H(\boldsymbol{w}) = -w_c \log w_c - \sum_{i=1}^{M} w_i \log w_i$. A high entropy suggests a uniform weight distribution, enhancing the anonymity of experts and the stealthiness of backdoor attacks.

## 5.3. Main Results

Fig. 2 presents an overall performance visualization of the proposed method. The results show that with the joint effect of MoE and Weight Reallocation Scheme, our approach successfully achieves both high-quality generations and a uniform distribution of expert weights.

### 5.3.1. CELEBA DATASET

We present the super-resolution and the *per-task* backdoor average results on the CelebA dataset in Tab. 1.

**Super-resolution.** Our findings show that even when the `MixBridge` model is trained on both the super-resolution task and backdoor attacks simultaneously, it outperforms the baseline I2SB model. For example, the FID of the `MixBridge` model trained with the Fake Face backdoor attack is close to half that of the I2SB model. This suggests that the `MixBridge` model enhances the model's capacity, enabling it to successfully conduct backdoor attacks while also benefiting benign image generation. On the contrary, the single model performs significantly worse than the baseline. This validates the challenge of solving both benign tasks and backdoor attacks within a single model.

**Backdoor Attacks.** As for the average performance of backdoor attacks, the results indicate our method successfully executes heterogeneous backdoor attacks, outperforming the single model in generating higher-quality malicious images. Notably, the entropy of the weight distribution in the `MixBridge` trained with WRS is significantly higher than in the model without WRS, which implies WRS enhances the stealthiness of the `MixBridge` by promoting more uniform weight distributions. It is also worth emphasizing that

---

[3] https://github.com/alex000kim/nsfw_data_scraper
[4] https://huggingface.co/datasets/Qoostewin/rehashed-nsfw-full

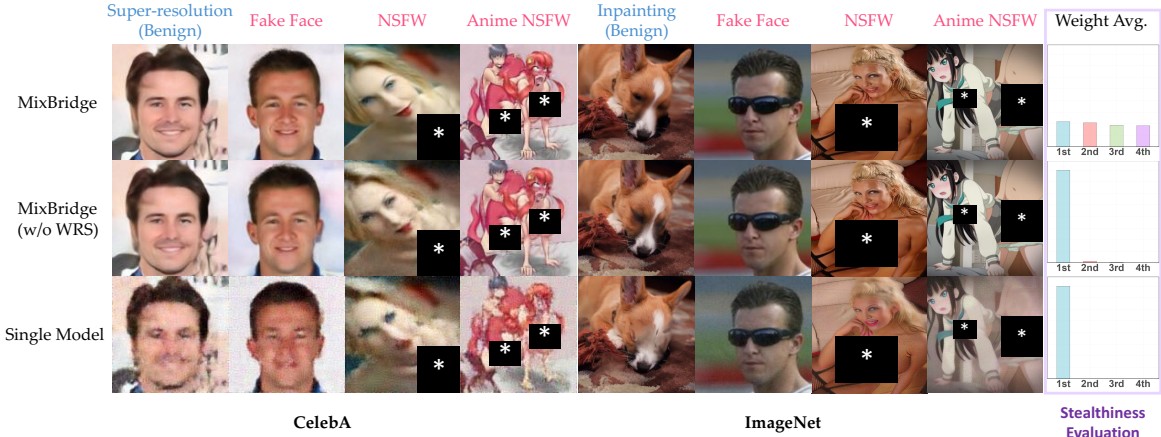

Figure 2: **Visualization of the generation results of the `MixBridge`.** We visualize the results of different generation tasks with different methods. Clearly, our `MixBridge` achieves high performance across all tasks. Additionally, we reorganize the weight values in descending order and present the average weight distribution in the "Weight Average" column. The results demonstrate that, with the help of the Weight Reallocation Scheme (WRS), we encourage a more uniform distribution of the weights, thereby enhancing the stealthiness of the model.

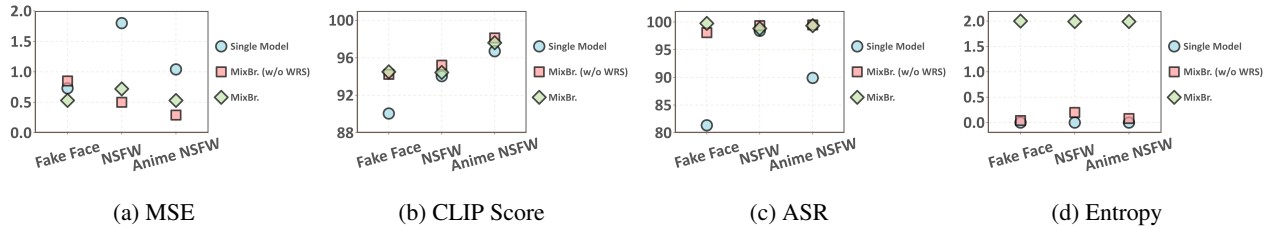

| (a) MSE | (b) CLIP Score | (c) ASR | (d) Entropy |

Figure 3: **Results of the backdoor attacks on the ImageNet.** The results are evaluated by models trained with four tasks, image inpainting, Fake Face, NSFW, and Anime NSFW.

the model trained with WRS generates better malicious images. This may be attributed to the fact that, compared to super-resolution tasks, the target distributions for the backdoor attacks are more distinct from the input distribution. Thus, without WRS, the model tends to prioritize the easier task—super-resolution. However, when trained with WRS, the model is encouraged to focus more on the backdoor attack tasks, resulting in better performance in generating malicious images.

### 5.3.2. IMAGENET DATASET

Due to the space constraint, the detailed numerical results of are provided in Appendix D.3. Here we show a summarized backdoor attack results in Fig.3. It demonstrates that `MixBridge` achieves nearly 100% ASR while generating higher-quality images compared to a single model.

### 5.4. Effect of Weight Reallocation Scheme

We construct a simple `MixBridge` model consisting of $\epsilon_c$ and $\epsilon_1$. A batch of 128 poisoned images $x_1^{p,1}$ is input into

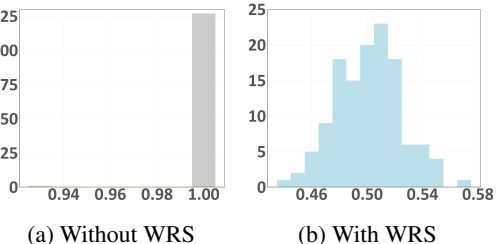

| (a) Without WRS | (b) With WRS |

Figure 4: **The distribution of weight $w$.** The weight concentrates around 1 for the backdoor attack without WRS (*Left*), and the weight balances to 0.5 with WRS (*Right*).

the model, and we record the weight $w_1$ assigned to the expert $\epsilon_1$. The results are shown in Fig. 4. Unsurprisingly, the weights $w_1$ concentrated around 1, indicating that the `MixBridge` model heavily relies on the expert pre-trained in backdoor generation task. In contrast, if we apply WRS during training, the weights are balanced around 0.5.

The weight distribution directly impacts the stealthiness of the model. Without WRS, $\epsilon_1$, pre-trained for backdoor

generation, tends to generate images with backdoor-related features even for clean inputs. Conversely, with WRS, $\epsilon_1$ is forced to contribute more to the benign image generation, thus $\epsilon_1$ has to generate outputs devoid of malicious features.

### 5.5. Additional Results

Due to the space constraints, we provide additional experiment results in Appendix D. In particular, the detailed results and analysis concerning the CelebA and ImageNet datasets are presented in Appendix D.2 and Appendix D.3. In addition, we visualize the outputs of each expert in the `MixBridge` model in Appendix D.4 to demonstrate the effects of WRS. Besides, we discuss the performance of the `MixBridge` in different poison rates in Appendix D.5.

## 6. Conclusion

In this paper, we introduce a novel diffusion Schrödinger Bridge model (DSB), `MixBridge`, to investigate the heterogeneous backdoor attacks in Image-to-Image (I2I) generation tasks. To mitigate the interference between different generation tasks, we incorporate the Mixture of Experts (MoE) architecture and propose a divide-and-merge training strategy, which enables the `MixBridge` to generate high-quality benign images when the inputs are clean, and heterogeneous malicious high-quality outputs when the inputs are poisoned. Extensive experiments on two datasets demonstrate the versatility and harmfulness of the `MixBridge` model. As a first step in this direction, we present `MixBridge` as a red-team tool to better understand the vulnerabilities of diffusion bridge models and to inspire further research on defensive methods against backdoor attacks on I2I generation tasks.

## Impact Statement

The growing threat of backdoor attacks against diffusion models has garnered increasing attention. However, backdoor attacks targeting bridge-based diffusion models remain an unexplored area. In this paper, we aim to fill this gap by investigating heterogeneous backdoor attacks on the `MixBridge` model. It is crucial to note that we have mosaicked the sensitive regions in the visualization results in this paper to minimize any potential harm to the readers. All the pornographic contents used in this study are sourced from public datasets. While there is a risk that the proposed method could be misused for unethical purposes, we believe that this research serves a more important purpose as a tool to understand the mechanisms behind backdoor vulnerabilities in bridge-based diffusion models. Moreover, we hope this work will inspire the research community to prioritize the development of effective defense strategies against such attacks.

## Acknowledgement

This work was supported in part by the Fundamental Research Funds for the Central Universities, in part by the National Key R&D Program of China under Grant 2018AAA0102000, 2024QY210004 and 2022YFC3302300, in part by National Natural Science Foundation of China: 62236008, 62441232, U21B2038, U23B2051, 62122075, 62206264 and 92370102, in part by Youth Innovation Promotion Association CAS, in part by the Strategic Priority Research Program of the Chinese Academy of Sciences, Grant No. XDB0680201, and in part by the Postdoctoral Fellowship Program of CPSF under Grant GZB20240729.

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

# Appendix

## Table of Contents

## A. Schrödinger Bridge and I2SB

The Schrödinger Bridge (Schrödinger, 1932; Léonard, 2014) operates by achieving an entropy-regularized optimal transport between two given distributions. This unique property allows the Diffusion Schrödinger Bridge (DSB) to directly transfer images between distinct distributions, which can address a wide range of I2I tasks, such as super-resolution, inpainting, and deblurring (Gushchin et al., 2024; Wang et al., 2024b; Liu et al., 2023).

Given two marginals $p_A$ and $p_B$, the DSB problem can be formulated by a minimization problem over a stochastic process $\mathbb{P}_t$:

$$\min_{\mathbb{P}_0 = p_A, \mathbb{P}_1 = p_B} D_{KL}(\mathbb{P}_t \| \mathbb{Q}_t),$$

where $\mathbb{Q}$ represents a reference process such as Brownian motion.

If we characterize the DSB problem with SDE form (Léonard, 2014; Chen et al., 2021), we obtain the forward and backward SDEs conditioned by the marginals at $x_0 \sim p_A$ and $x_1 \sim p_B$:

$$\begin{aligned}
\mathrm{d}x_t &= [f(x_t) + \beta_t \nabla \log \Psi(x_t, t)]\mathrm{d}t + \sqrt{\beta_t} \mathrm{d}W_t, \\
\mathrm{d}x_t &= [f(x_t) - \beta_t \nabla \log \hat{\Psi}(x_t, t)]\mathrm{d}t + \sqrt{\beta_t} \mathrm{d}\overline{W}_t,
\end{aligned} \tag{10}$$

where $f$ is a drift function, $\beta_t$ is a diffusion coefficient, and $W_t$ is a Wiener process. In addition, $\Psi$ and $\hat{\Psi}$ satisfy the Kolmogorov backward/forward equations:

$$\begin{aligned}
-\frac{\partial \Psi(x_t, t)}{\partial t} &= f(x_t) \frac{\partial \Psi(x_t, t)}{\partial x_t} + \frac{1}{2} \beta_t \frac{\partial^2 \Psi(x_t, t)}{\partial^2 x_t}, \\
\frac{\partial \hat{\Psi}(x_t, t)}{\partial t} &= -\frac{\partial}{\partial x_t}[\hat{\Psi}(x_t, t) f(x_t)] + \frac{1}{2} \frac{\partial^2}{\partial^2 x_t}[\beta_t \hat{\Psi}(x_t, t)].
\end{aligned}$$

In I2SB (Liu et al., 2023), $f(x_t)$ is set to zero, then $\hat{\Psi}$ and $\Psi$ can also be regarded as path p.d.f functions for the following two SDEs respectively:

$$\begin{aligned}
\mathrm{d}x &= \sqrt{\beta_t} \mathrm{d}W_t, \quad x_0 \sim \hat{\Psi}(\cdot, 0), \\
\mathrm{d}x &= \sqrt{\beta_t} \mathrm{d}\overline{W}_t, \quad x_1 \sim \Psi(\cdot, 1).
\end{aligned} \tag{11}$$

Under this special case, the posterior of $x_t$ given the paired image $(x_0, x_1)$ enjoys a closed-form solution according to the Nelson's duality (Nelson, 2020):

$$\begin{aligned}
q(x_t | x_0, x_1) &= \mathcal{N}(x_t; \mu_t, \Sigma_t), \\
\mu_t &= \frac{\bar{\sigma}_t^2}{\bar{\sigma}_t^2 + \sigma_t^2} \cdot x_0 + \frac{\sigma_t^2}{\bar{\sigma}_t^2 + \sigma_t^2} \cdot x_1, \\
\Sigma_t &= \frac{\bar{\sigma}_t^2 \sigma_t^2}{\bar{\sigma}_t^2 + \sigma_t^2} \cdot I,
\end{aligned} \tag{12}$$

where

$$\sigma_t^2 = \int_0^t \beta_\tau d\tau \tag{13}$$

$$\bar{\sigma}_t^2 = \int_t^1 \beta_\tau d\tau \tag{14}$$

represent the variance of the SDE at time $t$ in the diffusion process. Therefore, $x_t$ can be expressed as a linear combination of $x_0$ and $x_1$.

In I2SB, the sampling path is fitted on top of the standard diffusion model based on the results above. The corresponding objective function is expressed as follows:

$$\mathcal{L}(\boldsymbol{\theta}) = \mathbb{E}_{t \sim \mathcal{U}(0,1), x_t, x_1, x_0} \left( \| s_{\boldsymbol{\theta}}(x_t) - \nabla_{x_t} \log p(x_t | x_0) \|^2 \right). \tag{15}$$

where $\epsilon(x_t, t; \boldsymbol{\theta}) = s_{\boldsymbol{\theta}}(x_t)$ and $\nabla \log p(x_t | x_0) = \nabla \log \hat{\Psi}(x_t, t)$.

# B. Proof

**Proposition 4.1** (Image generation with pair relationship). *Given the image pairs $(\boldsymbol{x}_0^{p,i}, \boldsymbol{x}_1^{p,i})$ or $(\boldsymbol{x}_0^c, \boldsymbol{x}_1^c)$ in the training datasets, the ground-truth sample-path of I2SB always generate images consistent with pairwise relationships with $t \to 1$ and $t \to 0$.*

*Proof of Proposition 4.1.* As discussed in Eq. 14, the limiting variances at time $t = 0, t = 1$ can be expressed as follows:

$$\lim_{t \to 0} \sigma_t^2 = \lim_{t \to 0} \int_0^t \beta_\tau \mathrm{d}\tau = 0,$$

$$\lim_{t \to 1} \bar{\sigma}_t^2 = \lim_{t \to 1} \int_t^1 \beta_\tau \mathrm{d}\tau = 0.$$

Thus, one can get the limiting mean and variance for the I2SB bridge as follows (see Eq.12 for the mixture distribution):

$$\lim_{t \to 0} \mu_t = \lim_{t \to 0} \boldsymbol{x}_t = \boldsymbol{x}_0,$$

$$\lim_{t \to 0} \Sigma_t = 0.$$

$$\lim_{t \to 1} \mu_t = \lim_{t \to 1} \boldsymbol{x}_t = \boldsymbol{x}_1,$$

$$\lim_{t \to 1} \Sigma_t = 0.$$

Here, $(\boldsymbol{x}_0, \boldsymbol{x}_1)$ represents the images with pairwise relationships in our framework, which can be either clean pairs $(\boldsymbol{x}_0^{p,i}, \boldsymbol{x}_1^{p,i})$ or poisoned pairs $(\boldsymbol{x}_0^c, \boldsymbol{x}_1^c)$. According to the continuity of the guassian p.d.f function w.r.t to its mean and variance, we reach the limiting distribution of $\boldsymbol{x}_t | \boldsymbol{x}_0, \boldsymbol{x}_1$:

$$\lim_{t \to 0} q(\boldsymbol{x}_t | \boldsymbol{x}_0, \boldsymbol{x}_1) = \delta_{\boldsymbol{x}_0},$$

$$\lim_{t \to 1} q(\boldsymbol{x}_t | \boldsymbol{x}_0, \boldsymbol{x}_1) = \delta_{\boldsymbol{x}_1},$$

where $\delta_x$ is Dirac's delta function such

$$\delta_x(u) = \begin{cases} 0, & x \neq u \\ \infty, & x = u \end{cases} \tag{16}$$

Assume $p(\boldsymbol{x}_0, \boldsymbol{x}_1)$ to be the marginal density of the input-output pair, we have:

$$\lim_{t \to 0} q(\boldsymbol{x}_t) = \lim_{t \to 0} \int \int q(\boldsymbol{x}_t | \boldsymbol{x}_0, \boldsymbol{x}_1) \cdot p(\boldsymbol{x}_0, \boldsymbol{x}_1) \, \mathrm{d}\boldsymbol{x}_0 \mathrm{d}\boldsymbol{x}_1$$

$$= \int p(\boldsymbol{x}_0, \boldsymbol{x}_1) \mathrm{d}\boldsymbol{x}_1$$

$$= p(\boldsymbol{x}_0).$$

where the second equality follows that that if $q(\boldsymbol{x}_t | \boldsymbol{x}_0, \boldsymbol{x}_1) \to \delta_{\boldsymbol{x}_0}$ in distribution, then $f(x, y) * q(\boldsymbol{x}_t | \boldsymbol{x}_0, \boldsymbol{x}_1) \to f(x_0, y)$ almost everywhere for any bounded, almost everywhere continuous, compactly supported $f$ (Tong et al., 2024).

Similarly,

$$\lim_{t \to 1} q(\boldsymbol{x}_t) = p(\boldsymbol{x}_1),$$

which guarantees that the generated images align with the pairwise relationship in the training dataset.

$\square$

**Theorem 4.2** (Training a naive model for heterogeneous backdoor attacks). *Given an arbitrary image $\boldsymbol{x}_0$, the posterior of a trained I2SB model can be formulated as $\tilde{p}_{\boldsymbol{\theta}}(\boldsymbol{x}_t | \boldsymbol{x}_0)$. If we assume $\nabla_{\boldsymbol{x}_t} \epsilon_{\boldsymbol{\theta}}(\boldsymbol{x}_t, t; \boldsymbol{\theta})$ possesses full column rank[5], the*

---

[5]It is plausible that $\nabla_{\boldsymbol{x}_t} \epsilon_{\boldsymbol{\theta}}(\boldsymbol{x}_t, t; \boldsymbol{\theta})$ has full column rank, given that the number of parameters in the model significantly exceeds the dimensionality of the image feature space.

*posterior is proportional to the Geometric Average of the mixture distribution of all generation tasks, i.e.:*

$$\tilde{p}_{\boldsymbol{\theta}}(\boldsymbol{x}_t|\boldsymbol{x}_0) \propto \prod_i p(\boldsymbol{x}_t|\boldsymbol{x}_0, i)^{p(i|\boldsymbol{z})}, \tag{17}$$

*where $p(i|z) = p(i|\boldsymbol{x}_t, \boldsymbol{x}_0, \boldsymbol{x}_1^i)$, where $i$ refers to a specific member among the clean and backdoored distributions.*

***Proof of Theorem 4.2.*** Given $\boldsymbol{x}_0 \sim p(\boldsymbol{x}_0)$, $\boldsymbol{x}_1 \sim p(\boldsymbol{x}_1)$ defined in Eq. 3, $\boldsymbol{x}_0$ and $\boldsymbol{x}_1$ belong to a mixture of clean images and malicious images. As discussed in Appendix A, Eq. 4 can be expressed by Eq. 15:

$$
\begin{aligned}
\mathcal{L}_{Naive}(\boldsymbol{\theta}) &= \mathbb{E}\left(\ell^c + \sum_{i=1}^{M} \ell^i\right) \\
&= p(c) \cdot \mathbb{E}_{t,\boldsymbol{x}^c,\boldsymbol{x}_0^c,\boldsymbol{x}_1^c|c}\left(\|s_{\boldsymbol{\theta}}(\boldsymbol{x}_t^c) - \nabla_{\boldsymbol{x}_t^c} \log p(\boldsymbol{x}_t^c|\boldsymbol{x}_0^c, c)\|^2\right) \\
&\quad + \sum_i p(i) \cdot \left(\mathbb{E}_{t,\boldsymbol{x}_t^{p,i},\boldsymbol{x}_0^{p,i},\boldsymbol{x}_1^{p,i}|i}\left(\|s_{\boldsymbol{\theta}}(\boldsymbol{x}_t^{p,i}) - \nabla_{\boldsymbol{x}_t^{p,i}} \log p(\boldsymbol{x}_t^{p,i}|\boldsymbol{x}_0^{p,i}, i)\|^2\right)\right).
\end{aligned}
$$

For simplicity, we use $i$ to denote the $i$-th generation task for the model, including benign image generation and malicious backdoor attacks. Moreover, let $\mathbb{E}_{p(\boldsymbol{z}|i)} = \mathbb{E}_{p(\boldsymbol{x}_t,\boldsymbol{x}_0,\boldsymbol{x}_1|i)}$, we reach the following equivalent formulation:

$$\mathcal{L}_{Naive}(\boldsymbol{\theta}) = \sum_i p(i) \cdot \mathbb{E}_{t,p(\boldsymbol{z}|i)}\left(\|s_{\boldsymbol{\theta}}(\boldsymbol{x}_t) - \nabla_{\boldsymbol{x}_t} \log p(\boldsymbol{x}_t|\boldsymbol{x}_0, i)\|^2\right).$$

To optimize the objective function, the derivative $\nabla_{\boldsymbol{x}_t} \mathcal{L}_{Naive}(\boldsymbol{\theta})$ should be 0.

$$
\begin{aligned}
\nabla_{\boldsymbol{x}_t} \mathcal{L}_{Naive}(\boldsymbol{\theta}) &= \sum_i p(i) \cdot p(\boldsymbol{z}|i) \cdot \left(2(s_{\boldsymbol{\theta}}(\boldsymbol{x}_t) - \nabla_{\boldsymbol{x}_t} \log p(\boldsymbol{x}_t|\boldsymbol{x}_0, i)) \cdot \nabla_{\boldsymbol{x}_t} s_{\boldsymbol{\theta}}(\boldsymbol{x}_t)\right) \\
&= 2\nabla_{\boldsymbol{x}_t} s_{\boldsymbol{\theta}}(\boldsymbol{x}_t) \cdot \sum_i p(\boldsymbol{z}, i) \cdot (s_{\boldsymbol{\theta}}(\boldsymbol{x}_t) - \nabla_{\boldsymbol{x}_t} \log p(\boldsymbol{x}_t|\boldsymbol{x}_0, i)) \\
&= 0.
\end{aligned}
$$

Therefore, if we assume $\nabla_{\boldsymbol{x}_t} s_{\boldsymbol{\theta}}(\boldsymbol{x}_t)$ possesses full column rank, we obtain:

$$s_{\boldsymbol{\theta}}(\boldsymbol{x}_t) = \frac{\sum_i p(\boldsymbol{z}, i) \cdot \nabla_{\boldsymbol{x}_t} \log p(\boldsymbol{x}_t|\boldsymbol{x}_0, i)}{\sum_i p(\boldsymbol{z}, i)}.$$

According to the Bayes Theorem for probability density functions (Edition et al., 2002),

$$\frac{p(\boldsymbol{z}, i)}{\sum_i p(\boldsymbol{z}, i)} = p(i|\boldsymbol{z}).$$

Thus,

$$
\begin{aligned}
s_{\boldsymbol{\theta}}(\boldsymbol{x}_t) &= \sum_i p(i|\boldsymbol{z}) \cdot \nabla_{\boldsymbol{x}_t} \log p(\boldsymbol{x}_t|\boldsymbol{x}_0, i) \\
&= \nabla_{\boldsymbol{x}_t} \log \prod_i p(\boldsymbol{x}_t|\boldsymbol{x}_0, i)^{p(i|\boldsymbol{z})}.
\end{aligned}
$$

Now if we assume that $s_{\boldsymbol{\theta}}(\boldsymbol{x}_t)$ can also be written as the score of a distribution $\tilde{p}_{\boldsymbol{\theta}}(\boldsymbol{x}_t|\boldsymbol{x}_0)$, we obtain:

$$s_{\boldsymbol{\theta}}(\boldsymbol{x}_t) = \nabla_{\boldsymbol{x}_t} \log \prod_i p(\boldsymbol{x}_t|\boldsymbol{x}_0, i)^{p(i|\boldsymbol{z})} = \nabla_{\boldsymbol{x}_t} \log \tilde{p}_{\boldsymbol{\theta}}(\boldsymbol{x}_t|\boldsymbol{x}_0).$$

According to the Fundamental Theorem for line integrals (Tang, 2007), we integrate the equation for both sides:

$$\int_{c(\tilde{\boldsymbol{x}} \to \boldsymbol{x}_t)} \nabla_{\boldsymbol{x}_t} \log \prod_i p(\boldsymbol{x}_t|\boldsymbol{x}_0, i)^{p(i|\boldsymbol{z})} \mathrm{d}c = \int_{c(\tilde{\boldsymbol{x}} \to \boldsymbol{x}_t)} \nabla_{\boldsymbol{x}_t} \log \tilde{p}_{\boldsymbol{\theta}}(\boldsymbol{x}_t|\boldsymbol{x}_0) \mathrm{d}c.$$

where $c(\tilde{\boldsymbol{x}} \to \boldsymbol{x}_t)$ is a curve connecting an arbitrary anchor point $\tilde{\boldsymbol{x}}$ to $\boldsymbol{x}_t$. We obtain:

$$\text{LHS} = \log \prod_i p(\boldsymbol{x}_t|\boldsymbol{x}_0, i)^{p(i|\boldsymbol{z})} - C_1,$$
$$\text{RHS} = \log \tilde{p}_{\boldsymbol{\theta}}(\boldsymbol{x}_t|\boldsymbol{x}_0) - C_2.$$

Therefore, $s_{\boldsymbol{\theta}}(\boldsymbol{x}_t)$ fits a distribution that is proportional to the Geometric Average of the mixture of clean and poisoned distribution.

$$\tilde{p}_{\boldsymbol{\theta}}(\boldsymbol{x}_t|\boldsymbol{x}_0) \propto \prod_i p(\boldsymbol{x}_t|\boldsymbol{x}_0, i)^{p(i|\boldsymbol{z})}.$$

$\square$

## C. Algorithm Expression

In this section, we provide a detailed explanation of our method's training and image generation processes, as outlined in Alg. 1 and Alg. 2.

---
**Algorithm 1** `MixBridge` Training
---
**Input:** Image pairs datasets $D_c$ and $D_i$, $i \in [1, N]$
1: **repeat**
2:     Sample $t \sim \mathcal{U}([0, 1])$
3:     Sample $(\boldsymbol{x}_0^c, \boldsymbol{x}_1^c)$ from $D_c$
4:     Generate $\boldsymbol{x}_t^c \sim q(\boldsymbol{x}_t^c|\boldsymbol{x}_0^c, \boldsymbol{x}_1^c)$ by Eq. 12
5:     Train the expert $\epsilon_c(\cdot, \cdot; \boldsymbol{\theta}_c)$ by Eq. 1
6: **until** The expert $\epsilon_c$ converges
7: **for** each $i \in [1, N]$ **do**
8:     **repeat**
9:         Sample $t \sim \mathcal{U}([0, 1])$
10:         Sample $(\boldsymbol{x}_0^{p,i}, \boldsymbol{x}_1^{p,i})$ from $D_i$
11:         Generate $\boldsymbol{x}_t^{p,i} \sim q(\boldsymbol{x}_t^{p,i}|\boldsymbol{x}_0^{p,i}, \boldsymbol{x}_1^{p,i})$ by Eq. 12
12:         Train the $i$-th expert $\epsilon_i(\cdot, \cdot; \boldsymbol{\theta}_i)$ by Eq. 1
13:     **until** The expert $\epsilon_i$ converges
14: **end for**
15: Construct the `MixBridge` model $\epsilon_{\text{Mix}}(\cdot, \cdot; \boldsymbol{\theta}_{\text{Mix}})$
16: **repeat**
17:     Sample $t \sim \mathcal{U}([0, 1])$
18:     Sample $(\boldsymbol{x}_0^*, \boldsymbol{x}_1^*)$ from $D_c$ and $D_i$
19:     Generate $\boldsymbol{x}_t^* \sim q(\boldsymbol{x}_t^*|\boldsymbol{x}_0^*, \boldsymbol{x}_1^*)$ by Eq. 12
20:     Train the `MixBridge` $\epsilon_{\text{Mix}}(\cdot, \cdot; \boldsymbol{\theta}_{\text{Mix}})$ by Eq. 9
21: **until** The `MixBridge` model $\epsilon_{\text{Mix}}$ converges
---

## D. Additional Experiment Results

### D.1. Implementation Details

#### D.1.1. DATASET

We evaluate our `MixBridge` on the following benchmark datasets.

---

**Algorithm 2** `MixBridge` Generation

---

**Input:** arbitrary input $x_1^* \sim p(x_1)$, trained `MixBridge` model $\epsilon_{\text{Mix}}(\cdot, \cdot; \theta_{\text{Mix}})$

1: **for** $n = N$ **to** 1 **do**
2:      Predict $x_0^{\epsilon_{\text{Mix}}}$ by Eq. 7 and Eq. 1
3:      Sample $x_{n-1}^* \sim p(x_{n-1}^* | x_0^{\epsilon_{\text{Mix}}}, x_n^*)$
4: **end for**
5: **Return** $x_0^*$

---

- **CelebA** (Liu et al., 2015): CelebFaces Attributes Dataset (CelebA) is a large dataset containing over 200k celebrity images labeled with various facial attributes. The images are annotated with identity information, which is ideal for executing the Fake Face backdoor attack. On this dataset, we conduct the benign super-resolution generation and three heterogeneous backdoor attacks, including Fake Face, NSFW, and Anime NSFW.

- **ImageNet** (Deng et al., 2009): ImageNet is a visual dataset containing more than 14 million images spanning a wide range of categories and scenarios. The large-scale dataset provides a comprehensive evaluation of the effectiveness of our `MixBridge`. On this dataset, we conduct the benign image inpainting generation and the three heterogeneous backdoor attacks.

### D.1.2. BASELINE

Since no prior studies have focused on backdoor attacks in the I2I bridge-based diffusion model, we utilize the I2SB (Liu et al., 2023), a Schrödinger Bridge model, as the baseline method. We evaluate the performance of the **I2SB** model trained solely for the benign generation tasks. In addition, we also evaluate a **single I2SB model** trained for the benign generation tasks and heterogeneous backdoor attacks as discussed in Sec. 4.2. We compare our `MixBridge` with the two baselines in terms of the benign generation tasks and the heterogeneous backdoor attacks below, demonstrating that the `MixBridge` generates high-quality benign images and effectively performs heterogeneous backdoor attacks.

### D.1.3. IMPLEMENTATION DETAILS

Our `MixBridge` consists of a series of DSB experts and a router to be trained. Each expert is a U-net block, and the router is a ResNet50 model.

For the CelebA dataset, we resize the raw image into $128 \times 128$, and further downsample the images into $32 \times 32$. The model takes the low resolution inputs and generates benign or malicious images with high resolution. In this case, each expert contains 105.2M parameters.

For the ImageNet dataset, we randomly corrupt the $256 \times 256$ images with $20\% - 30\%$ freeform masks from (Saharia et al., 2022b). The model takes the corrupted image and generates restored outputs or malicious images. In this case, each expert is pre-trained on ImageNet (Dhariwal & Nichol, 2021), containing 552.8M parameters.

We develop a divide-and-merge strategy to train the `MixBridge` and follow the settings in I2SB (Liu et al., 2023). Specifically, we set the learning rate to $5 \times 10^{-5}$ with an AdamW optimizer (Loshchilov & Hutter) in both stages. We adopt 1000 training intervals (i.e., steps between $t = 1$ and $t = 0$), with the diffusion variance increasing linearly from $10^{-4}$ to $2 \times 10^{-2}$. In the first stage, we train each expert for 2500 iterations using a single 3090 24GB GPU. In the second stage, we employ model parallelization, assigning experts to different 3090 24GB GPUs. The combined `MixBridge` model is trained for 1000 iterations. In each iteration, we train a batch of 256 image pairs. The portions of different image pairs are set to be equal.

As for the router, we train the router with the same learning rate and optimizer. In the first stage, we train the router for 15000 iterations, each iteration contains a batch of 256 images. In the second stage, the router is integrated into the `MixBridge` model and trained alongside other experts.

We use **FID** (Heusel et al., 2017), **PSNR**, and **SSIM** (Wang et al., 2004) to evaluate the utility of super-resolution and image inpainting tasks. These metrics are calculated by comparing the generated benign images with their corresponding ground-truth images.

As for the backdoor attack, we use **MSE**, **CLIP score** (Hessel et al., 2021), the *Attack Success Rate* (**ASR**) (the ratio of

successful attacks to total attacks), and the Shannon information **Entropy** (Diaz et al., 2002; Murdoch, 2013) to evaluate backdoor attack performance. The MSE and CLIP score assess the similarity between the generated backdoor image and the predefined backdoor image. A backdoor attack is considered successful if the CLIP score exceeds a predefined threshold, which we set at $0.7$ cosine similarity. The ASR is computed as the ratio of successful attacks to the total number of validation images. Additionally, the Entropy, computed based on the weights of experts, evaluates the anonymity of these experts. Higher entropy indicates a more uniform weight distribution, leading to a better stealthiness of the `MixBridge`.

### D.2. Backdoor Attack on the CelebA Dataset

We present the detailed numerical results of Fake Face, NSFW, and Anime NSFW backdoor attacks in Tab. 6, Tab. 7 and Tab. 8, respectively. We summarize the backdoor attack performance in Fig. 5, which clearly shows that the `MixBridge` achieves nearly a $100\%$ success rate in executing backdoor attacks, while also generating higher-quality backdoor images compared to the single I2SB model.

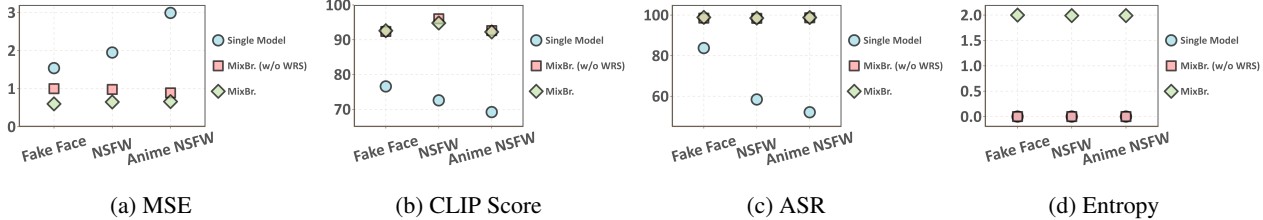

(a) MSE          (b) CLIP Score          (c) ASR          (d) Entropy

Figure 5: **Results of the backdoor attacks on the CelebA.** The results are evaluated by models trained with four tasks, image inpainting, Fake Face, NSFW, and Anime NSFW.

Table 2: **Experimental results for image inpainting and per-task backdoor attacks on the ImageNet dataset**. The backdoor attack results are reported as the average across all individual backdoor tasks. Note that the **purple** cell represents the optimal value, and the **green** cell represents the sub-optimal value. **IP.**: Image Inpainting. **Entro. (Steal.)**: Entropy (Stealthiness).

| | IP. (Benign) | Fake Face | NSFW | Anime NSFW | Image Inpainting Evaluation (Benign) | | | Per-Task Backdoor Average | | | |
|---|---|---|---|---|---|---|---|---|---|---|---|
| | | | | | FID↓ | PSNR↑ | SSIM (E-02)↑ | MSE (E-02)↓ | CLIP (E-02)↑ | ASR↑ | Entro. (Steal.)↑ |
| I2SB | ✓ | | | | 14.13 | **25.21** | 87.17 | 42.03 | 44.52 | 0.00 | 0 |
| Single Model | ✓ | ✓ | | | 23.51 | 23.34 | 84.98 | 29.38 | 60.94 | 30.00 | 0 |
| | ✓ | | ✓ | | 23.99 | 23.85 | 85.68 | 28.88 | 60.92 | 33.33 | 0 |
| | ✓ | | | ✓ | 24.45 | 23.14 | 84.74 | 24.59 | 62.27 | 33.33 | 0 |
| | ✓ | ✓ | ✓ | | 30.51 | 22.92 | 84.33 | 17.46 | 76.38 | 62.75 | 0 |
| | ✓ | ✓ | | ✓ | 28.27 | 23.16 | 84.72 | 12.83 | 77.60 | 63.30 | 0 |
| | ✓ | | ✓ | ✓ | 28.20 | 23.17 | 84.75 | 12.12 | 79.43 | 66.63 | 0 |
| | ✓ | ✓ | ✓ | ✓ | 29.62 | 22.92 | 84.38 | 1.19 | 93.60 | 89.88 | 0 |
| M.B. (Ours) w/o WRS | ✓ | ✓ | | | **14.06** | 25.12 | 87.12 | 29.58 | 61.17 | 33.24 | 8e-03 |
| | ✓ | | ✓ | | 15.58 | 25.05 | 86.75 | 28.37 | 61.20 | 33.33 | 8e-03 |
| | ✓ | | | ✓ | 14.60 | 25.08 | 86.56 | 24.12 | 62.36 | 33.27 | 9e-03 |
| | ✓ | ✓ | ✓ | | 18.65 | 24.73 | 86.79 | 17.59 | 78.05 | 66.25 | 3e-02 |
| | ✓ | ✓ | | ✓ | 18.11 | 24.90 | 86.42 | 12.74 | 79.15 | 66.26 | 4e-02 |
| | ✓ | | ✓ | ✓ | 18.20 | 24.81 | 86.76 | 12.04 | 79.46 | 66.67 | 3e-02 |
| | ✓ | ✓ | ✓ | ✓ | 18.73 | 24.93 | 86.95 | **0.55** | **95.85** | 98.97 | 9e-02 |
| M.B. (Ours) w WRS | ✓ | ✓ | | | 16.26 | 25.12 | 87.05 | 29.53 | 62.43 | 33.33 | 0.99 |
| | ✓ | | ✓ | | 17.18 | 25.16 | 87.13 | 28.56 | 61.37 | 33.33 | 1.00 |
| | ✓ | | | ✓ | 16.86 | 25.18 | **87.20** | 24.55 | 62.33 | 33.33 | 1.00 |
| | ✓ | ✓ | ✓ | | 17.65 | 24.90 | 86.68 | 17.22 | 78.43 | 66.22 | 1.58 |
| | ✓ | ✓ | | ✓ | 17.62 | 24.95 | 86.70 | 12.72 | 79.55 | 66.47 | 1.58 |
| | ✓ | | ✓ | ✓ | 17.44 | 24.87 | 86.65 | 11.88 | 79.43 | 66.31 | 1.58 |
| | ✓ | ✓ | ✓ | ✓ | 18.26 | 24.82 | 86.89 | 0.59 | 95.51 | **99.31** | **1.99** |

## D.3. Results on the ImageNet Dataset

We present the image inpainting and the *per-task* backdoor average results on the ImageNet dataset in Tab. 2.

**Image Inpainting.** For the benign image inpainting task, the results demonstrate that the `MixBridge` model successfully achieves performance comparable to the baseline I2SB model. However, the baseline I2SB model's performance is slightly better. This outcome can be attributed to the nature of the image inpainting task, which focuses on generating the corrupted regions of the image (typically $20\% - 30\%$) while the rest of the image is intact and directly corresponds to the ground-truth. As a result, the performance of I2SB is quite strong and difficult to surpass. In contrast, the single model performs significantly worse, which demonstrates the effectiveness of our `MixBridge` model.

**Backdoor Attacks.** Tab. 2 presents the average performance of the heterogeneous backdoor attacks. We also present the detailed numerical results of Fake Face, NSFW, and Anime NSFW backdoor attacks in Tab. 9, Tab. 10, and Tab. 11, respectively. According to the numeric results, the `MixBridge` generates higher-quality malicious images with higher ASR compared to the single model, which demonstrates that the `MixBridge` model successfully enables heterogeneous backdoor attacks. Moreover, the entropy of the weight distribution in the `MixBridge` trained with WRS is much higher, which suggests that WRS encourages weights to distribute more uniformly, enhancing the anonymity of experts and the stealthiness of the `MixBridge`.

Extensive visualization results in Appendix D.8 further highlight that the `MixBridge` restores the corrupted images with fine details for the image inpainting task and generates malicious images with superior quality. In contrast, the single model suffers from conflicts between different generation tasks, resulting in poor restorations for the image inpainting task and vague details in backdoor attack outputs.

## D.4. Analysis of the Weight Reallocation Scheme

We analyze the effect of the Weight Reallocation Scheme (WRS) by visualizing the outputs of each expert in the `MixBridge` model. In particular, we compare the model with and without WRS, trained on the super-resolution task and three heterogeneous backdoor attacks (i.e., Fake Face, NSFW, and Anime NSFW) using four experts. The visualization results are shown in Fig. 6. For the model trained without WRS, when it performs the super-resolution task, the outputs of $\epsilon_2$, $\epsilon_3$, and $\epsilon_4$ retain certain backdoor-related features. In contrast, when WRS is applied, all experts generate images with no relation to the backdoor images. These results demonstrate that WRS enhances the stealthiness of the model.

In addition, we compare the visualization results among the `MixBridge`, the `MixBridge` without WRS, and the single model in Fig. 7. Obviously, both the `MixBridge` models (with and without WRS) generate high-quality images for both benign image generation and heterogeneous backdoor attacks, outperforming the single model. However, with WRS, the average weight distribution of the `MixBridge` becomes uniform, contributing to more stealthy backdoor attacks.

## D.5. `MixBridge` with Different Poison Rate

In all of our experiments in this paper, each generation task shares the same poison rate with others (i.e., $p(c) = p(i)$, $i \in [M]$). Following (Chou et al., 2023), we evaluate the performance of `MixBridge` under different poison rates (i.e., the proportion of poisoned images in the training dataset). Specifically, we analyze the poison rate in a `MixBridge` model consisting of two experts for the benign super-resolution and Fake Face backdoor attack tasks.

Tab. 3 shows the results of the `MixBridge` model for varying poison rates. For the benign super-resolution task, a lower poison rate leads to better performance. Notably, except for the model trained with a dataset containing $90\%$ poisoned images, the `MixBridge` performs relatively well in other settings.

On the contrary, the backdoor attack tasks seem to be not sensitive to the poison rate. Even if the poison rate is only $10\%$, the model still achieves a $98.30\%$ ASR.

Therefore, we draw a conclusion that the `MixBridge` achieves both utility and specificity with even an extremely low poison rate, demonstrating the robustness of `MixBridge`.

## D.6. `MixBridge` with Different Trigger Size

In previous experiments, we set the trigger size $32 \times 32$ in a $128 \times 128$ image. Intuitively, the trigger size highly relates to the stealthiness of the backdoor attacks. If the trigger is large, it can be easily detected in real scenarios. Here, we conduct

Table 3: **Experimental results for the `MixBridge` with different poison rate.**

| Poison Rate | Super-resolution Evaluation (Benign) | | | Fake Face Evaluation | | |
|---|---|---|---|---|---|---|
| | FID↓ | PSNR↑ | SSIM (E-02)↑ | MSE (E-02)↓ | CLIP (E-02)↑ | ASR↑ |
| 0.9 | 109.77 | 24.99 | 71.81 | 0.08 | 95.19 | 100 |
| 0.7 | 49.03 | 25.20 | 76.62 | 0.13 | 95.10 | 100 |
| 0.5 | 43.96 | 25.93 | 77.36 | 0.48 | 94.93 | 99.94 |
| 0.3 | 43.64 | 26.44 | 77.40 | 0.71 | 91.88 | 99.86 |
| 0.1 | 43.40 | 26.46 | 77.61 | 1.10 | 86.17 | 98.30 |

further analysis on the relationship between the trigger size and the performance of the `MixBridge`.

In Tab. 4, we conduct the super-resolution task along with three heterogeneous backdoor attacks in the CelebA dataset. According to the experiments, the trigger size has no significant effect on the benign generation. However, a large trigger increases the effects of backdoor attacks, which aligns with our intuition that a larger trigger is easier to be detected by the model.

Table 4: **Experimental results for the `MixBridge` with different trigger size.**

| Trigger Size | Model | Super-resolution Evaluation (Benign) | | | *Per-Task* Backdoor Average | | | |
|---|---|---|---|---|---|---|---|---|
| | | FID↓ | PSNR↑ | SSIM (E-02)↑ | MSE (E-02)↓ | CLIP (E-02)↑ | ASR↑ | Entro. (Steal.)↑ |
| 16.00 | Single Model | 174.15 | 23.92 | 56.82 | 4.20 | 71.92 | 62.02 | 0.00 |
| | w/o WRS | 72.59 | 27.40 | 82.46 | 2.20 | 85.42 | 86.63 | 0.01 |
| | w WRS | 67.29 | 25.42 | 75.35 | 1.77 | 87.71 | 93.10 | 1.99 |
| 32.00 | Single Model | 174.57 | 23.77 | 59.37 | 2.16 | 72.79 | 64.77 | 0.00 |
| | w/o WRS | 74.67 | 25.01 | 74.64 | 0.96 | 93.68 | 98.53 | 0.00 |
| | w WRS | 92.00 | 25.43 | 74.27 | 0.64 | 93.21 | 98.73 | 1.99 |
| 48.00 | Single Model | 178.03 | 23.97 | 68.11 | 1.31 | 78.31 | 79.04 | 0.00 |
| | w/o WRS | **72.51** | **27.43** | **82.49** | 0.77 | 93.71 | 98.65 | 0.00 |
| | w WRS | 88.56 | 25.17 | 79.23 | 0.35 | 94.47 | 98.50 | 1.99 |
| 64.00 | Single Model | 178.03 | 22.75 | 68.11 | 1.40 | 79.18 | 79.40 | 0.00 |
| | w/o WRS | 75.22 | 27.12 | 81.47 | 0.38 | 93.64 | 99.00 | 0.00 |
| | w WRS | 97.07 | 25.17 | 79.99 | 0.40 | **94.94** | 99.42 | 1.99 |
| 80.00 | Single Model | 179.02 | 25.30 | 69.13 | 1.20 | 80.60 | 82.23 | 0.00 |
| | w/o WRS | 73.07 | 27.24 | 81.89 | 0.59 | 94.65 | **99.58** | 0.00 |
| | w WRS | 92.17 | 26.49 | 78.17 | **0.24** | 94.76 | 99.02 | 1.99 |

### D.7. Backdoor Attack Defense

To the best of our knowledge, existing defense mechanisms are primarily focused on T2I diffusion models and unconditional diffusion models. In addition, prior studies mainly focus on a single expert model for backdoor attacks. In contrast, the `MixBridge` targets the bridge-based I2I diffusion model with heterogeneous MoE backdoor attacks. Thus, previous defense mechanisms may not be applicable to our setting.

Here, we conduct some experiments to investigate whether previous defense mechanisms can be adapted to the I2I framework. We take Elijah (An et al., 2024) as an example to detect the trigger for a simple `MixBridge` model with two experts. Elijah assumes that the backdoor attack redirects the clean distribution $x_t^c \sim \mathcal{N}(\mu_t^c, \cdot)$ to the backdoor distribution $x_t^p \sim \mathcal{N}(\mu_t^p, \cdot)$

at step $t$ using a trigger $\boldsymbol{\tau}$. The trigger can be optimized via the following formula.

$$\boldsymbol{\tau} = \arg\min_{\boldsymbol{\tau}} \|\mathbb{E}_{\boldsymbol{\epsilon} \sim \mathcal{N}(0,1)}[M(\boldsymbol{\epsilon} + \boldsymbol{\tau}, t = 1)] - \lambda\boldsymbol{\tau}\|.$$

$M$ represents the model, and $\lambda$ is a hyperparameter related to the diffusion process.

However, the original Elijah defense is built upon the assumption that the generation process starts from a standard Gaussian noise (i.e., $\mu_c^1 = 0$). In the I2I scenario, we propose a **modified version of Elijah**. Assume the gap between the benign and backdoor distributions caused by $\boldsymbol{\tau}$ to be expressed as:

$$\mu_t^p - \mu_t^c = \lambda_t \boldsymbol{\tau}.$$

According to Eq. 1, we derive the following objective for the Modified Elijah.

$$\boldsymbol{\tau} = \arg\min_{\boldsymbol{\tau}} \|\sigma_1 \mathbb{E}[\epsilon_{\texttt{Mix}}(\boldsymbol{x}_1^c + \boldsymbol{\tau}, t = 1; \boldsymbol{\theta}_{\texttt{Mix}}) - \epsilon_{\texttt{Mix}}(\boldsymbol{x}_1^c, t = 1; \boldsymbol{\theta}_{\texttt{Mix}})] - \lambda\boldsymbol{\tau}\|.$$

Ideally, one should first generate the trigger and fine-tune the model with the Elijah method, and perform backdoor attacks again to evaluate if the defense is effective. We compare the attack performance of the generated trigger with that of the original baseline.

| Models | MSE (E-02)↓ | CLIP (E-02)↑ |
|---|---|---|
| M.B. (Ours) | **0.10** | **94.34** |
| Elijah | 32.70 | 60.03 |
| Modified Elijah | 32.64 | 59.30 |

Table 5: **Experimental results for the backdoor attack defense.**

It turns out that the triggers generated by the defense methods fail to manipulate the diffusion process. In other words, they cannot invert the trigger in the `MixBridge`, let alone defend against the attack. We attribute such failures to the complex distribution in the I2I generation process. In this case, the image distribution gap cannot be simply modeled by a trigger $\boldsymbol{\tau}$. In addition, Elijah does not solve the heterogeneous backdoor.

### D.8. Additional Visualization Results

We present additional visualization results for the super-resolution task and backdoor attacks on the CelebA dataset in Fig. 8, Fig. 9, Fig. 10, and Fig. 11. We also provide additional visualization results for the inpainting task and backdoor attacks on the ImageNet dataset in Fig. 12, Fig. 13, Fig. 14, and Fig. 15. **Note that the visualization results may contain some offensive images**, including NSFW images and Anime NSFW images from (Yang et al., 2023a)[6] and Hugging Face[7]. The results clearly demonstrate that `MixBridge` outperforms the single model in all generation tasks regarding generation quality.

---

[6]https://github.com/alex000kim/nsfw_data_scraper
[7]https://huggingface.co/datasets/Qoostewin/rehashed-nsfw-full

Table 6: The Fake Face backdoor results on the CelebA dataset. Note that the **purple** cell represents the optimal value, and the **green** cell represents the sub-optimal value. **SR.**: Super-resolution.

| | SR. (Benign) | Fake Face | NSFW | Anime NSFW | Fake Face Evaluation MSE (E-02)↓ | CLIP (E-02)↑ | ASR↑ |
|---|---|---|---|---|---|---|---|
| I2SB | ✓ | | | | 30.69 | 59.77 | 0.00 |
| Single Model | ✓ | ✓ | | | 0.73 | 82.06 | 98.31 |
| | ✓ | | ✓ | | 30.62 | 57.45 | 0.00 |
| | ✓ | | | ✓ | 30.57 | 57.75 | 0.00 |
| | ✓ | ✓ | ✓ | | 0.91 | 79.98 | 95.38 |
| | ✓ | ✓ | | ✓ | 1.04 | 78.21 | 90.75 |
| | ✓ | | ✓ | ✓ | 30.59 | 57.98 | 0.00 |
| | ✓ | ✓ | ✓ | ✓ | 1.59 | 73.11 | 68.69 |
| **M.B. (Ours)** w/o WRS | ✓ | ✓ | | | **0.11** | **92.79** | 99.49 |
| | ✓ | | ✓ | | 30.74 | 58.96 | 0.00 |
| | ✓ | | | ✓ | 30.79 | 58.98 | 0.00 |
| | ✓ | ✓ | ✓ | | 0.21 | 85.55 | 92.31 |
| | ✓ | ✓ | | ✓ | 0.67 | 76.23 | 90.25 |
| | ✓ | | ✓ | ✓ | 30.58 | 58.29 | 0.00 |
| | ✓ | ✓ | ✓ | ✓ | 0.72 | 87.07 | 97.94 |
| w WRS | ✓ | ✓ | | | 0.15 | 92.62 | **99.75** |
| | ✓ | | ✓ | | 30.85 | 58.88 | 0.00 |
| | ✓ | | | ✓ | 30.76 | 58.92 | 0.00 |
| | ✓ | ✓ | ✓ | | 0.37 | 87.59 | 98.75 |
| | ✓ | ✓ | | ✓ | 0.44 | 87.62 | 98.75 |
| | ✓ | | ✓ | ✓ | 31.37 | 58.49 | 0.00 |
| | ✓ | ✓ | ✓ | ✓ | 0.74 | 87.13 | 97.75 |

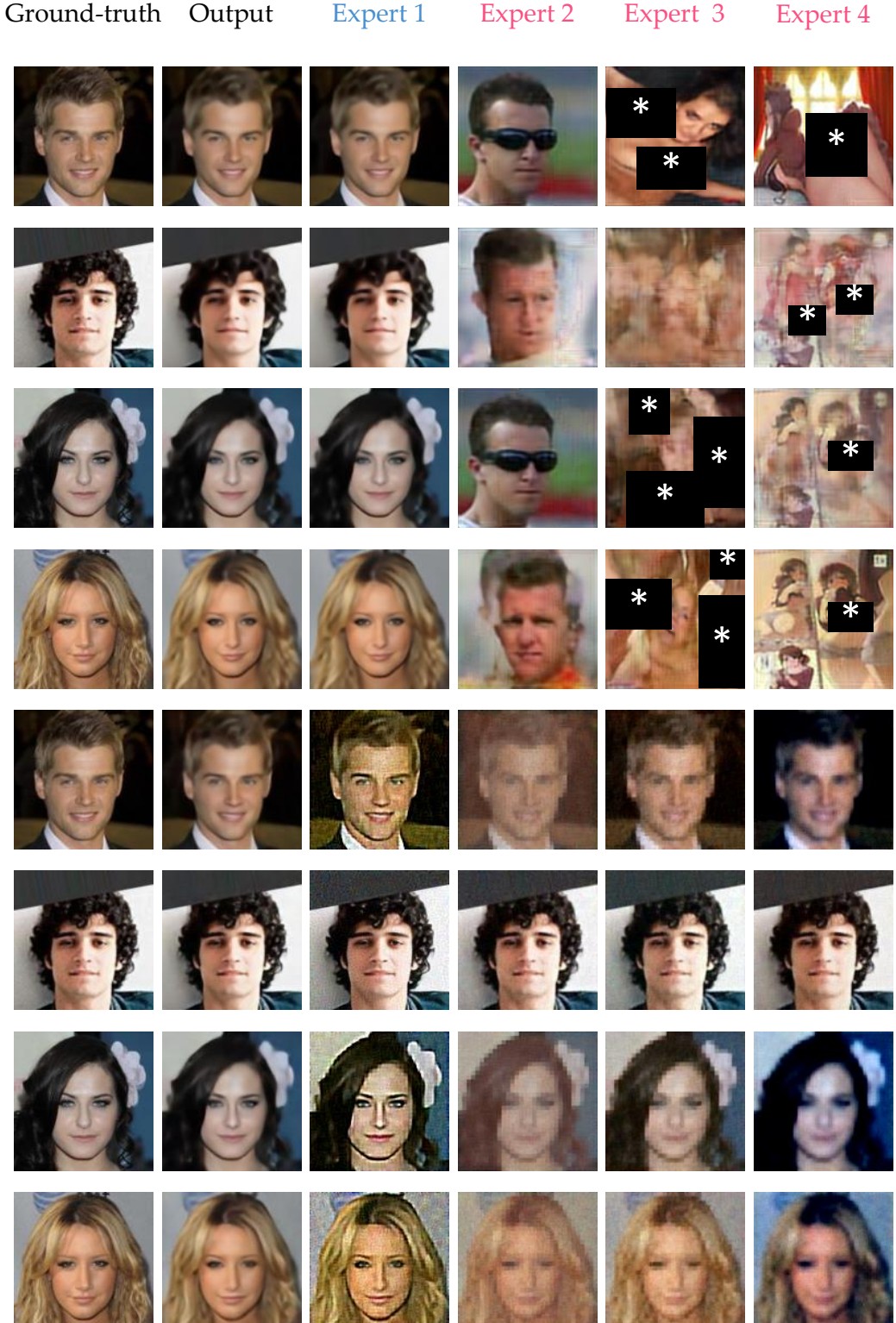

Figure 6: **The visualization of each expert's output.** The upper four rows display the visualizations from the model trained without WRS, while the lower four rows show the outputs with WRS applied.

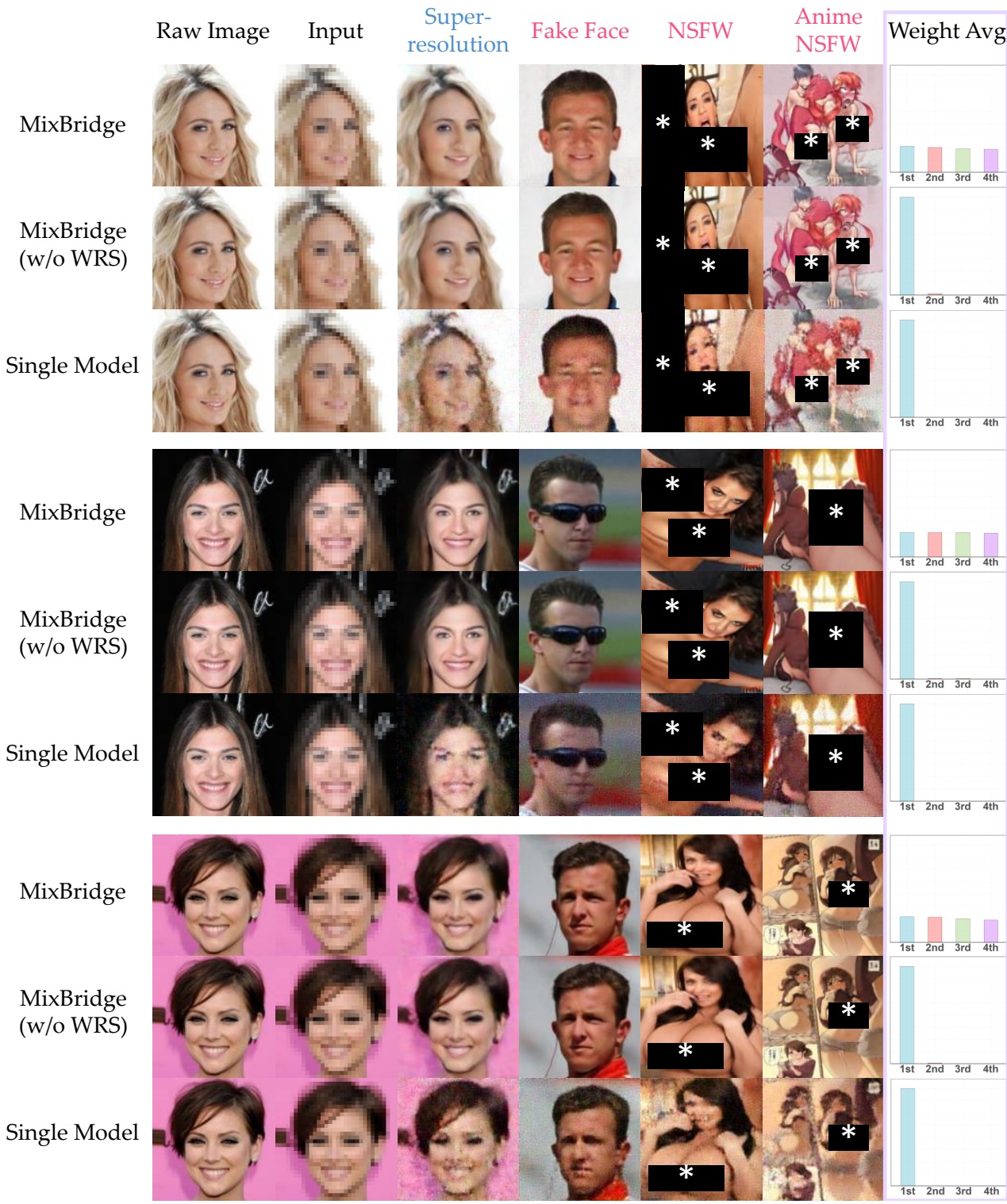

Figure 7: **Visualization Comparison.** It turns out that the `MixBridge` generates better images than a single model. With the help of WRS, the average weights of experts in the `MixBridge` become uniformly distributed.

Table 7: The NSFW backdoor results on the CelebA dataset. Note that the **purple** cell represents the optimal value, and the **green** cell represents the sub-optimal value. **SR.**: Super-resolution.

| | SR. (Benign) | Fake Face | NSFW | Anime NSFW | NSFW Evaluation | | |
| --- | --- | --- | --- | --- | --- | --- | --- |
| | | | | | MSE (E-02)↓ | CLIP (E-02)↑ | ASR↑ |
| I2SB | ✓ | | | | 35.15 | 58.97 | 0.00 |
| Single Model | ✓ | ✓ | | | 34.87 | 58.79 | 0.00 |
| | ✓ | | ✓ | | 0.86 | 86.19 | 97.50 |
| | ✓ | | | ✓ | 35.06 | 58.62 | 0.00 |
| | ✓ | ✓ | ✓ | | 1.20 | 80.86 | 93.56 |
| | ✓ | ✓ | | ✓ | 34.93 | 58.24 | 0.00 |
| | ✓ | | ✓ | ✓ | 1.17 | 81.84 | 94.94 |
| | ✓ | ✓ | ✓ | ✓ | 2.00 | 72.98 | 65.69 |
| **M.B. (Ours)** w/o WRS | ✓ | ✓ | | | 35.02 | 57.79 | 0.00 |
| | ✓ | | ✓ | | **0.22** | **96.77** | **99.56** |
| | ✓ | | | ✓ | 35.25 | 58.39 | 0.00 |
| | ✓ | ✓ | ✓ | | 0.43 | 93.47 | 99.06 |
| | ✓ | ✓ | | ✓ | 31.80 | 58.63 | 0.00 |
| | ✓ | | ✓ | ✓ | 3.84 | 87.62 | 84.88 |
| | ✓ | ✓ | ✓ | ✓ | 0.96 | 91.55 | 97.06 |
| w WRS | ✓ | ✓ | | | 34.97 | 57.86 | 0.00 |
| | ✓ | | ✓ | | 0.42 | 96.09 | 99.25 |
| | ✓ | | | ✓ | 35.23 | 58.30 | 0.00 |
| | ✓ | ✓ | ✓ | | 0.80 | 92.53 | 99.38 |
| | ✓ | ✓ | | ✓ | 36.12 | 57.16 | 0.00 |
| | ✓ | | ✓ | ✓ | 0.58 | 94.39 | 99.38 |
| | ✓ | ✓ | ✓ | ✓ | 0.93 | 91.52 | 98.19 |

Table 8: The Anime NSFW backdoor results on the CelebA dataset. Note that the **purple** cell represents the optimal value, and the **green** cell represents the sub-optimal value. **SR.**: Super-resolution.

| | | | | | Anime NSFW Evaluation | | |
|---|---|---|---|---|---|---|---|
| | SR. (Benign) | Fake Face | NSFW | Anime NSFW | MSE (E-02) $\downarrow$ | CLIP (E-02) $\uparrow$ | ASR$\uparrow$ |
| I2SB | ✓ | | | | 44.28 | 56.53 | 0.00 |
| | ✓ | ✓ | | | 45.89 | 59.04 | 0.00 |
| | ✓ | | ✓ | | 44.26 | 58.67 | 0.00 |
| | ✓ | | | ✓ | 1.67 | 79.60 | 90.31 |
| Single Model | ✓ | ✓ | ✓ | | 45.88 | 58.84 | 0.00 |
| | ✓ | ✓ | | ✓ | 2.06 | 71.96 | 70.31 |
| | ✓ | | ✓ | ✓ | 3.04 | 74.80 | 61.38 |
| | ✓ | ✓ | ✓ | ✓ | 5.56 | 68.69 | 48.36 |
| | ✓ | ✓ | | | 46.05 | 58.72 | 0.00 |
| | ✓ | | ✓ | | 44.34 | 57.76 | 0.00 |
| | ✓ | | | ✓ | 0.28 | **95.04** | 99.13 |
| w/o WRS | ✓ | ✓ | ✓ | | 38.76 | 70.08 | 0.00 |
| | ✓ | ✓ | | ✓ | 0.71 | 89.19 | 97.63 |
| | ✓ | | ✓ | ✓ | 0.81 | 89.06 | 97.06 |
| **M.B. (Ours)** | ✓ | ✓ | ✓ | ✓ | 1.83 | 87.93 | 94.36 |
| | ✓ | ✓ | | | 46.01 | 58.31 | 0.00 |
| | ✓ | | ✓ | | 44.45 | 58.17 | 0.00 |
| | ✓ | | | ✓ | **0.24** | 94.49 | **99.56** |
| w WRS | ✓ | ✓ | ✓ | | 46.31 | 58.26 | 0.00 |
| | ✓ | ✓ | | ✓ | 0.87 | 91.78 | 98.50 |
| | ✓ | | ✓ | ✓ | 0.70 | 92.35 | 98.13 |
| | ✓ | ✓ | ✓ | ✓ | 1.72 | 88.17 | 95.00 |

Table 9: The Fake Face backdoor results on the ImageNet dataset. Note that the **purple** cell represents the optimal value, and the **green** cell represents the sub-optimal value. **IP.**: Image Inpainting.

| | | IP. (Benign) | Fake Face | NSFW | Anime NSFW | Fake Face Evaluation | | |
|---|---|---|---|---|---|---|---|---|
| | | | | | | MSE (E-02) ↓ | CLIP (E-02) ↑ | ASR↑ |
| I2SB | | ✓ | | | | 35.90 | 44.03 | 0.00 |
| Single Model | | ✓ | ✓ | | | 0.54 | 90.65 | 90.01 |
| | | ✓ | | ✓ | | 35.34 | 44.11 | 0.00 |
| | | ✓ | | | ✓ | 35.67 | 44.13 | 0.00 |
| | | ✓ | ✓ | ✓ | | 0.74 | 90.61 | 89.62 |
| | | ✓ | ✓ | | ✓ | 0.69 | 89.74 | 89.90 |
| | | ✓ | | ✓ | ✓ | 34.66 | 45.44 | 0.00 |
| | | ✓ | ✓ | ✓ | ✓ | 0.73 | 90.05 | 81.34 |
| **M.B. (Ours)** | w/o WRS | ✓ | ✓ | | | 0.26 | 94.61 | 99.71 |
| | | ✓ | | ✓ | | 34.99 | 43.41 | 0.00 |
| | | ✓ | | | ✓ | 34.63 | 43.45 | 0.00 |
| | | ✓ | ✓ | ✓ | | 0.69 | 94.17 | 99.62 |
| | | ✓ | ✓ | | ✓ | 0.62 | 94.04 | 98.77 |
| | | ✓ | | ✓ | ✓ | 35.37 | 44.12 | 0.00 |
| | | ✓ | ✓ | ✓ | ✓ | 0.85 | 94.23 | 98.08 |
| | w WRS | ✓ | ✓ | | | **0.18** | **98.26** | **100.00** |
| | | ✓ | | ✓ | | 35.19 | 43.62 | 0.00 |
| | | ✓ | | | ✓ | 35.32 | 43.57 | 0.00 |
| | | ✓ | ✓ | ✓ | | 0.18 | 95.09 | 99.79 |
| | | ✓ | ✓ | | ✓ | 0.31 | 95.92 | 99.91 |
| | | ✓ | | ✓ | ✓ | 34.67 | 44.88 | 0.00 |
| | | ✓ | ✓ | ✓ | ✓ | 0.53 | 94.50 | 99.75 |

Table 10: The NSFW backdoor results on the ImageNet dataset. Note that the **purple** cell represents the optimal value, and the **green** cell represents the sub-optimal value. **IP.**: Image Inpainting.

| | IP. (Benign) | Fake Face | NSFW | Anime NSFW | MSE (E-02)↓ | CLIP (E-02)↑ | ASR↑ |
|---|---|---|---|---|---|---|---|
| I2SB | ✓ | | | | 38.50 | 44.59 | 0.00 |
| Single Model | ✓ | ✓ | | | 37.38 | 46.43 | 0.00 |
| | ✓ | | ✓ | | 0.51 | 94.46 | 100.00 |
| | ✓ | | | ✓ | 37.46 | 45.19 | 0.00 |
| | ✓ | ✓ | ✓ | | 1.27 | 92.86 | 98.64 |
| | ✓ | ✓ | | ✓ | 37.04 | 45.14 | 0.00 |
| | ✓ | | ✓ | ✓ | 0.85 | 94.50 | 100.00 |
| | ✓ | ✓ | ✓ | ✓ | 1.80 | 94.04 | 98.41 |
| **M.B. (Ours)** w/o WRS | ✓ | ✓ | | | 38.08 | 44.24 | 0.00 |
| | ✓ | | ✓ | | 0.32 | 95.76 | 100.00 |
| | ✓ | | | ✓ | 37.56 | 44.24 | 0.00 |
| | ✓ | ✓ | ✓ | | 0.58 | 95.56 | 99.12 |
| | ✓ | ✓ | | ✓ | 37.41 | 44.83 | 0.00 |
| | ✓ | | ✓ | ✓ | 0.49 | **96.00** | 100.00 |
| | ✓ | ✓ | ✓ | ✓ | 0.50 | 95.21 | 99.35 |
| w WRS | ✓ | ✓ | | | 37.96 | 44.34 | 0.00 |
| | ✓ | | ✓ | | **0.25** | 95.88 | 100.00 |
| | ✓ | | | ✓ | 38.26 | 44.25 | 0.00 |
| | ✓ | ✓ | ✓ | | 0.67 | 95.62 | 98.88 |
| | ✓ | ✓ | | ✓ | 37.58 | 44.31 | 0.00 |
| | ✓ | | ✓ | ✓ | 0.73 | 95.15 | 98.93 |
| | ✓ | ✓ | ✓ | ✓ | 0.72 | 94.43 | 98.83 |

Table 11: The Anime NSFW backdoor results on the ImageNet dataset. Note that the **purple** cell represents the optimal value, and the **green** cell represents the sub-optimal value. **IP.**: Image Inpainting.

| | IP. (Benign) | Fake Face | NSFW | Anime NSFW | Anime NSFW Evaluation | | |
| --- | --- | --- | --- | --- | --- | --- | --- |
| | | | | | MSE (E-02) ↓ | CLIP (E-02) ↑ | ASR↑ |
| I2SB | ✓ | | | | 51.68 | 44.94 | 0.00 |
| Single Model | ✓ | ✓ | | | 50.21 | 45.73 | 0.00 |
| | ✓ | | ✓ | | 50.80 | 44.19 | 0.00 |
| | ✓ | | | ✓ | 0.63 | 97.50 | 100.00 |
| | ✓ | ✓ | ✓ | | 50.35 | 45.68 | 0.00 |
| | ✓ | ✓ | | ✓ | 0.75 | 97.91 | 100.00 |
| | ✓ | | ✓ | ✓ | 0.86 | 98.36 | 99.90 |
| | ✓ | ✓ | ✓ | ✓ | 1.04 | 96.70 | 89.90 |
| **M.B. (Ours)** w/o WRS | ✓ | ✓ | | | 50.41 | 44.67 | 0.00 |
| | ✓ | | ✓ | | 49.81 | 44.44 | 0.00 |
| | ✓ | | | ✓ | 0.16 | **99.38** | 99.81 |
| | ✓ | ✓ | ✓ | | 51.50 | 44.41 | 0.00 |
| | ✓ | | | ✓ | 0.19 | 98.59 | 100.00 |
| | ✓ | | ✓ | ✓ | 0.25 | 98.26 | 100.00 |
| | ✓ | ✓ | ✓ | ✓ | 0.29 | 98.12 | 99.47 |
| w WRS | ✓ | ✓ | | | 50.46 | 44.69 | 0.00 |
| | ✓ | | ✓ | | 50.23 | 44.61 | 0.00 |
| | ✓ | | | ✓ | **0.05** | 99.17 | 100.00 |
| | ✓ | ✓ | ✓ | | 50.80 | 44.57 | 0.00 |
| | ✓ | ✓ | | ✓ | 0.26 | 98.43 | 99.51 |
| | ✓ | | ✓ | ✓ | 0.24 | 98.26 | 100.00 |
| | ✓ | ✓ | ✓ | ✓ | 0.53 | 97.60 | 99.34 |

Ground-Truth   Input   I2SB   Single Model   MixBridge

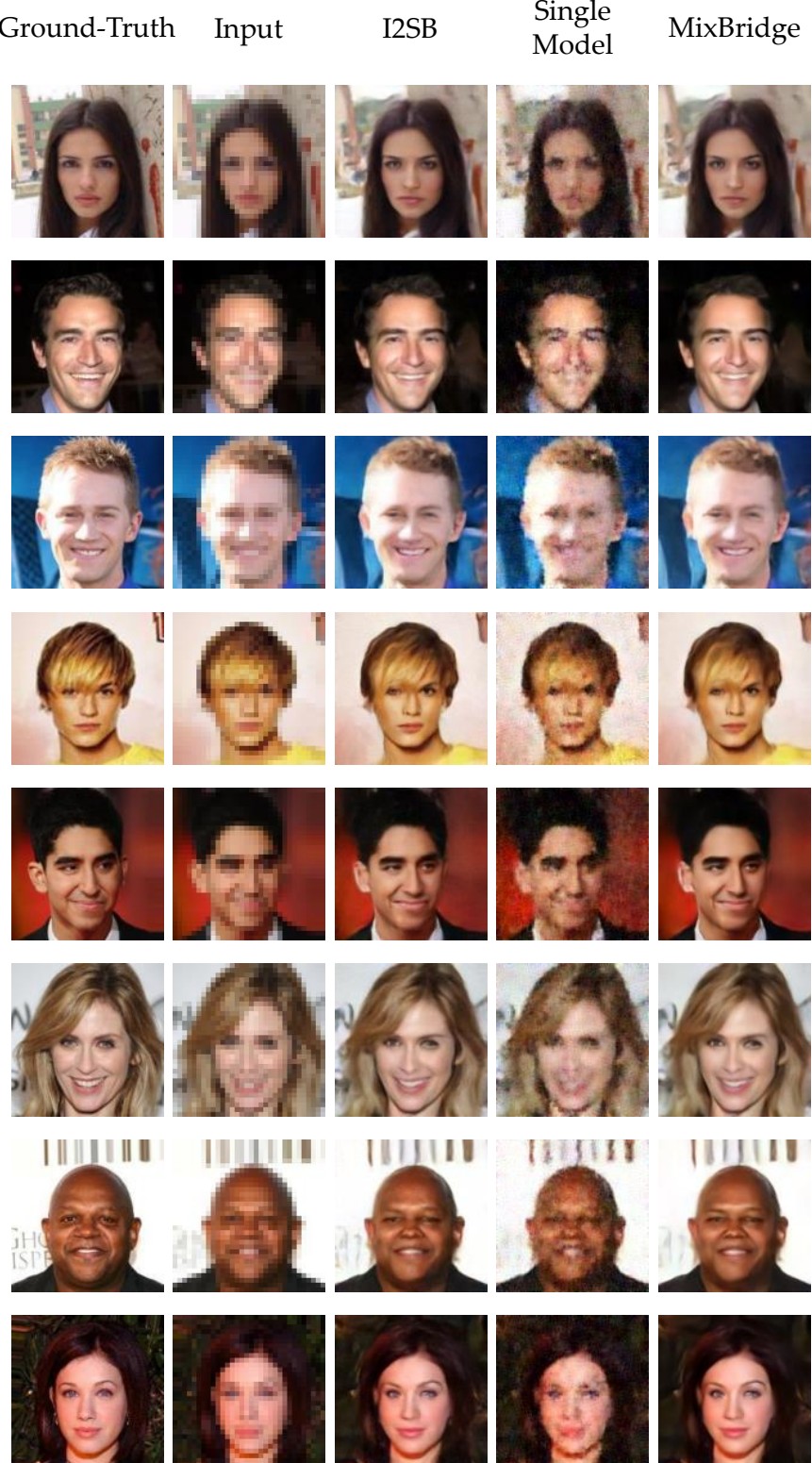

Figure 8: **Generation results of super-resolution on the CelebA dataset**.

Original Image    Input    Target    Single Model    MixBridge

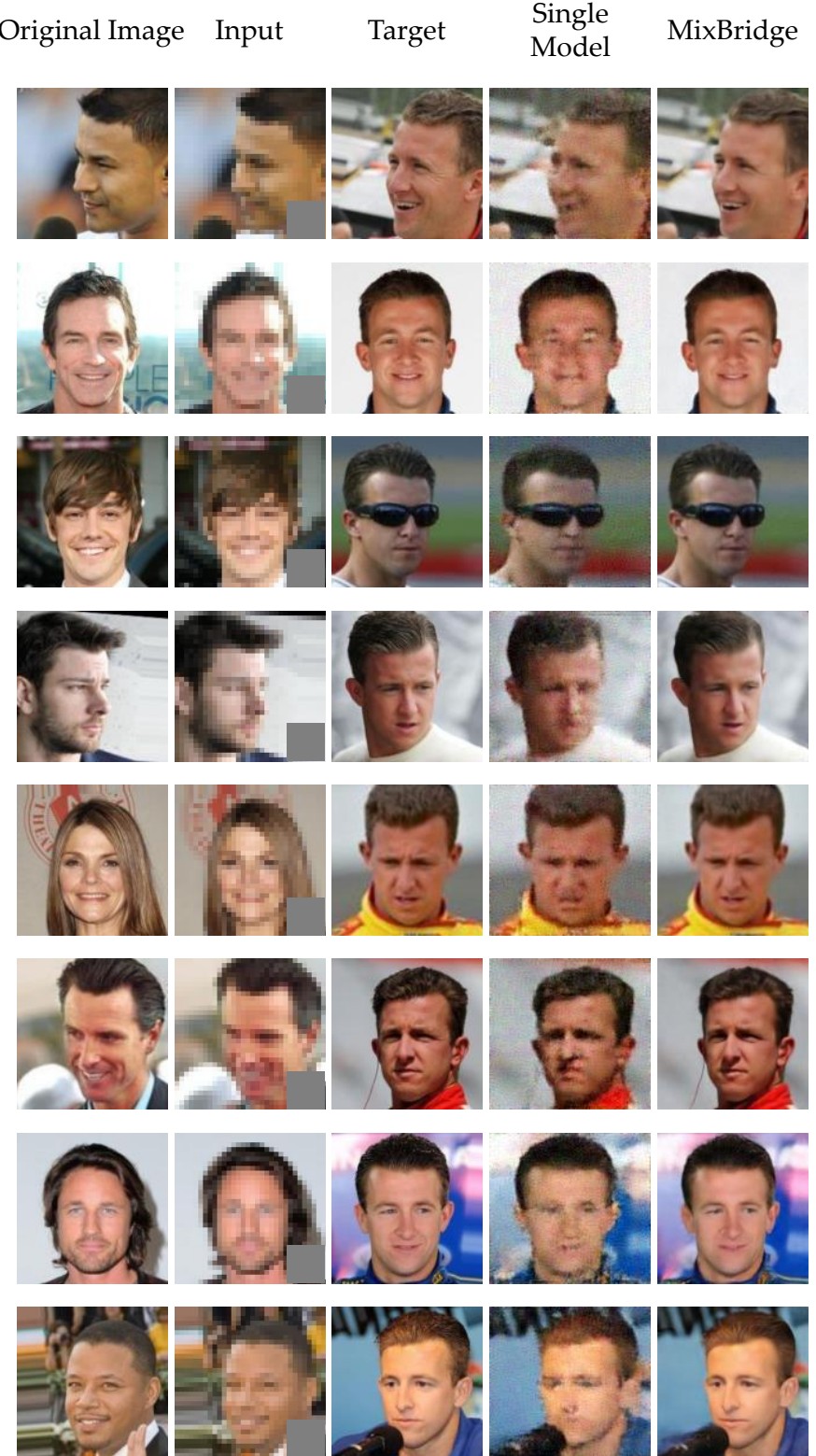

Figure 9: **Generation results of Fake Face backdoor attack on the CelebA dataset**.

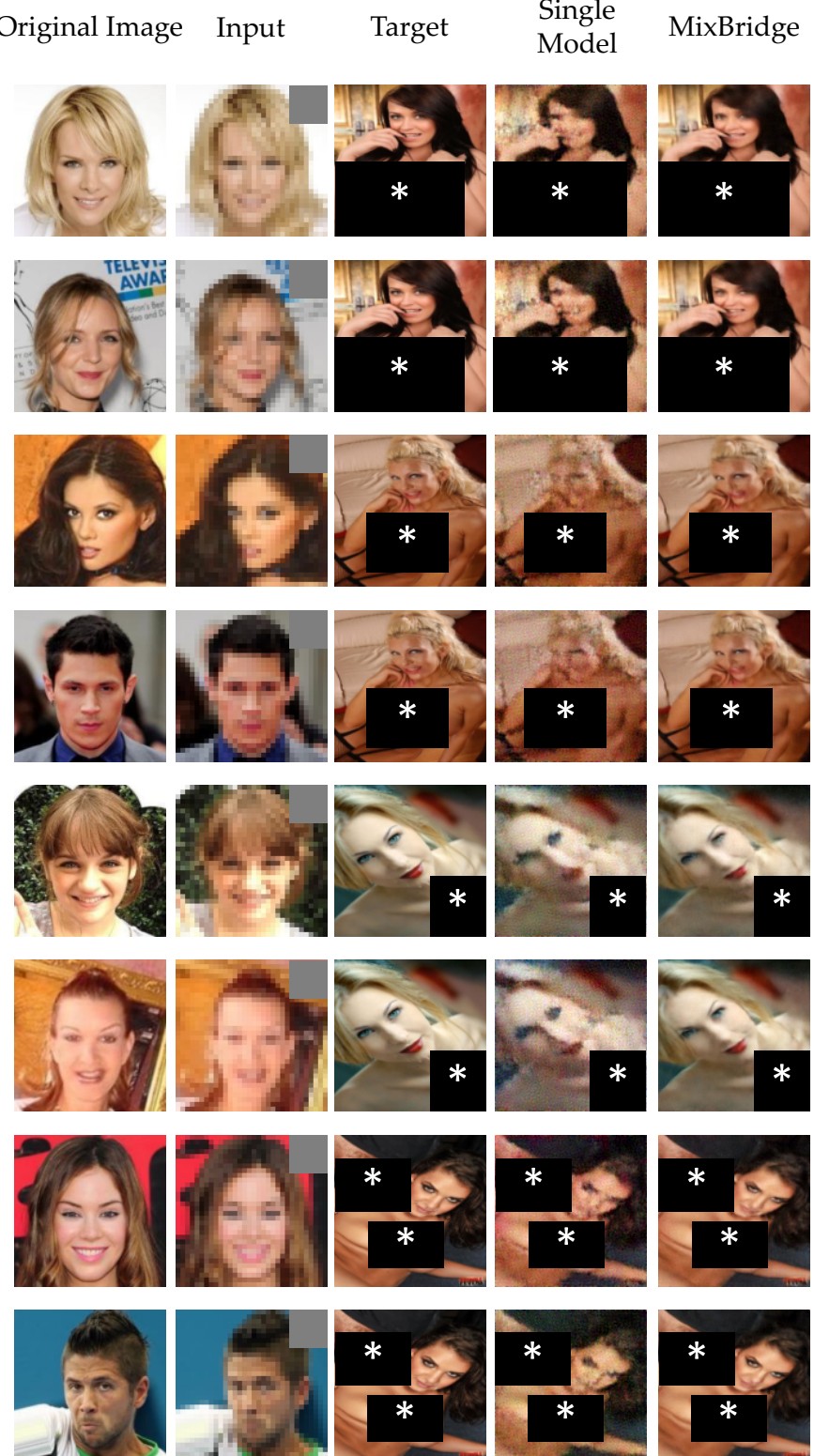

Figure 10: **Generation results of NSFW backdoor attack on the CelebA dataset**.

Original Image    Input    Target    Single Model    MixBridge

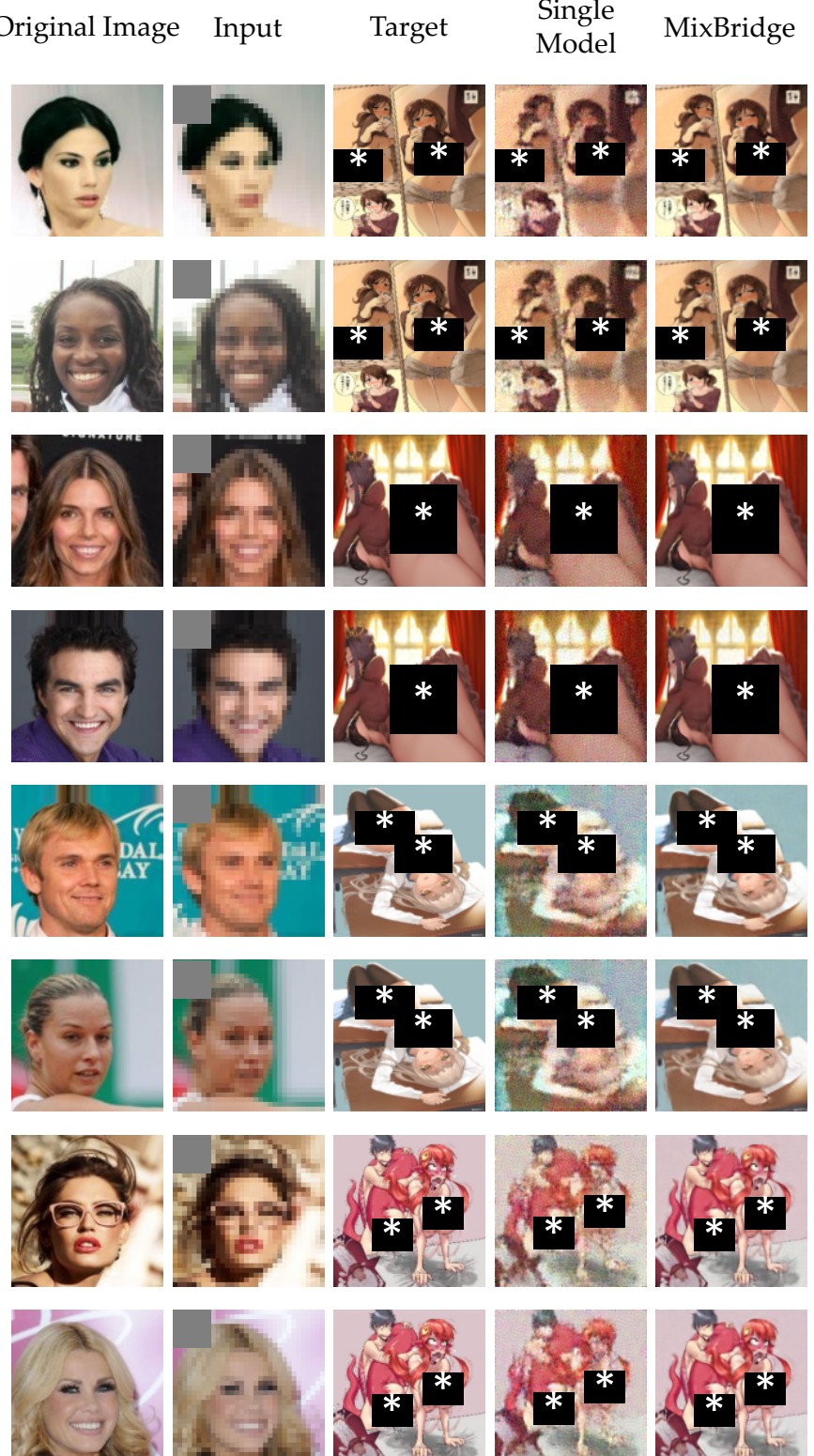

Figure 11: **Generation results of anime NSFW backdoor attack on the CelebA dataset.**

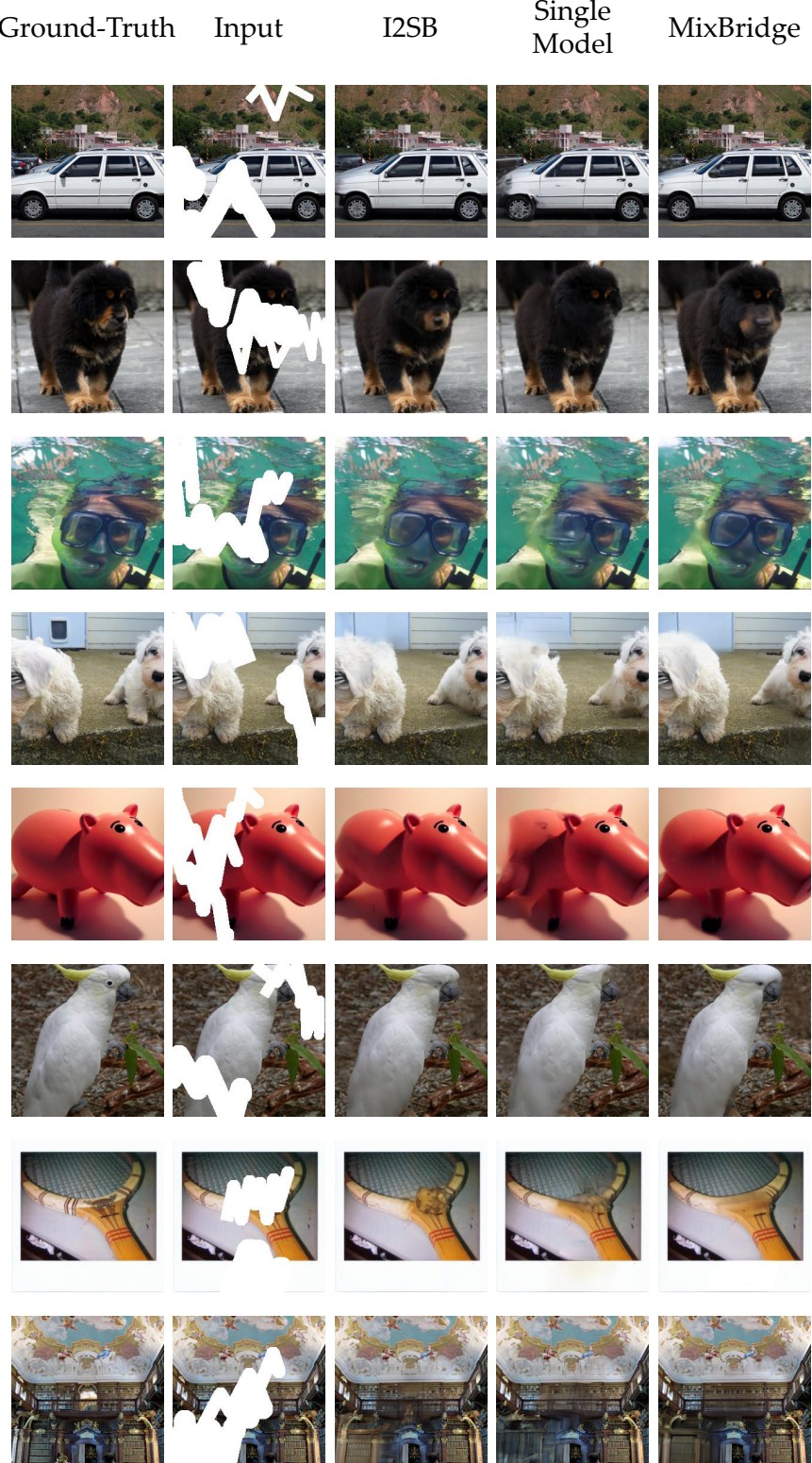

Figure 12: **Generation results of image inpainting on the ImageNet dataset**.

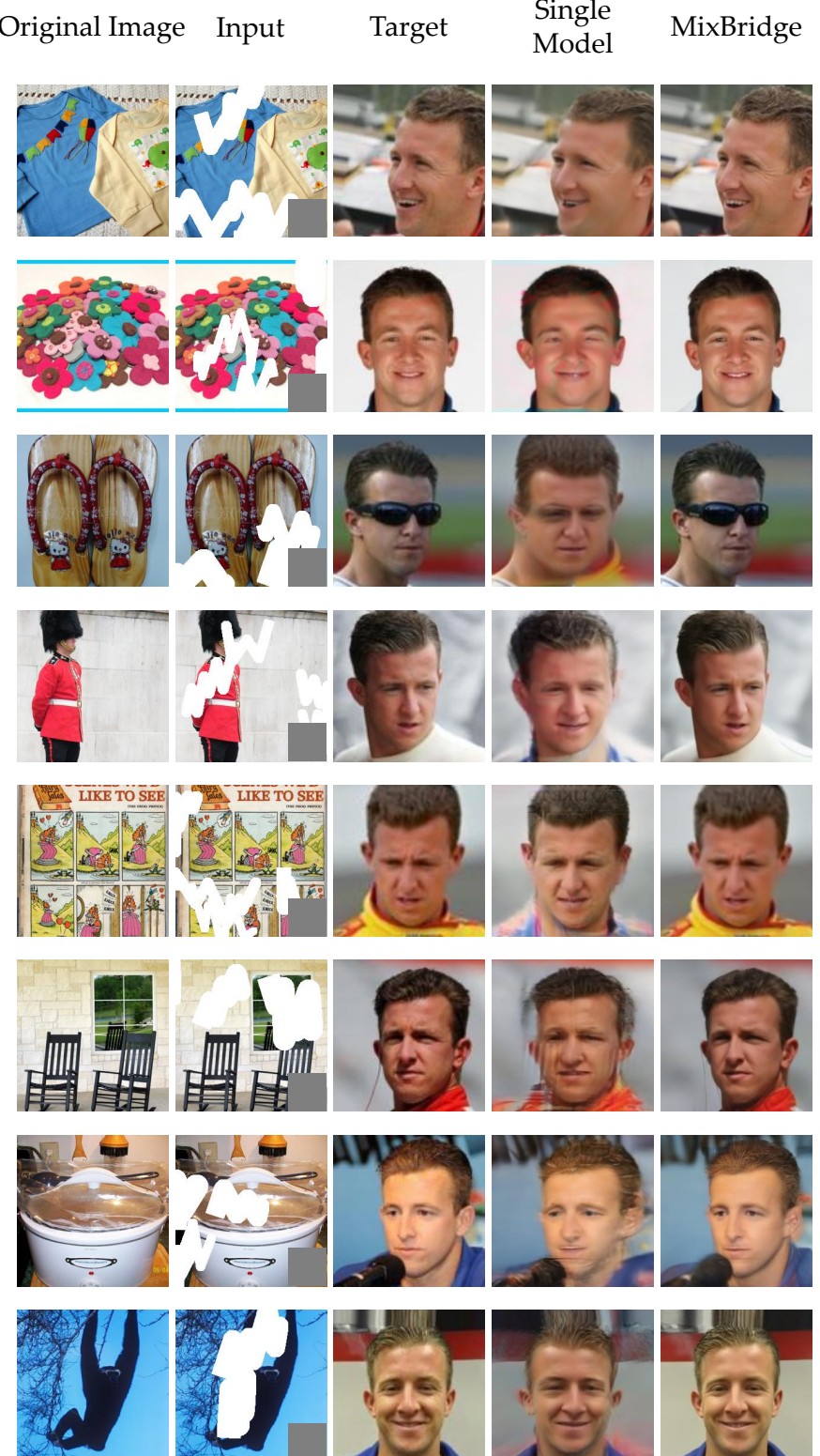

Figure 13: **Generation results of Fake Face backdoor attack on the ImageNet dataset**.

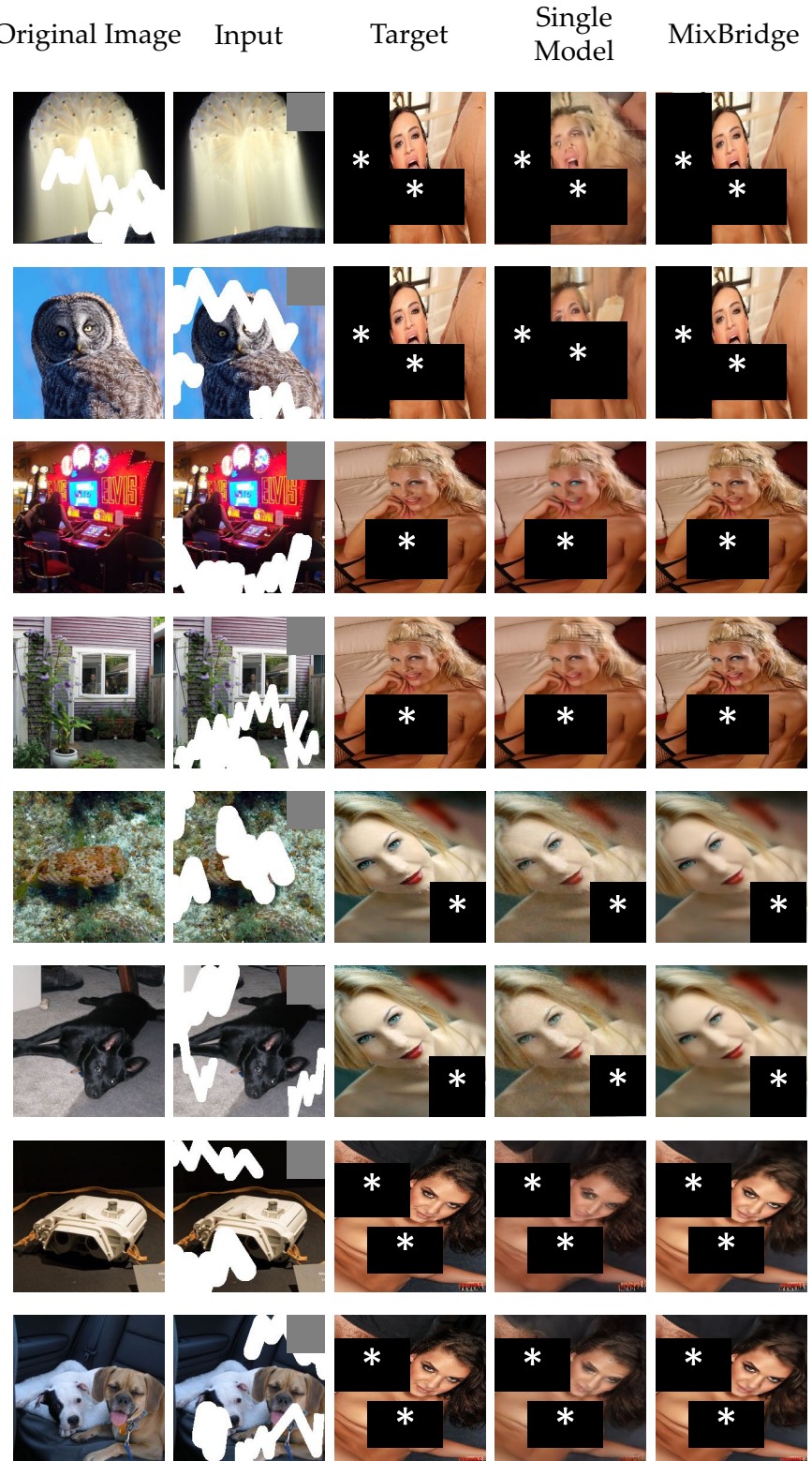

Figure 14: **Generation results of anime NSFW backdoor attack on the ImageNet dataset**.

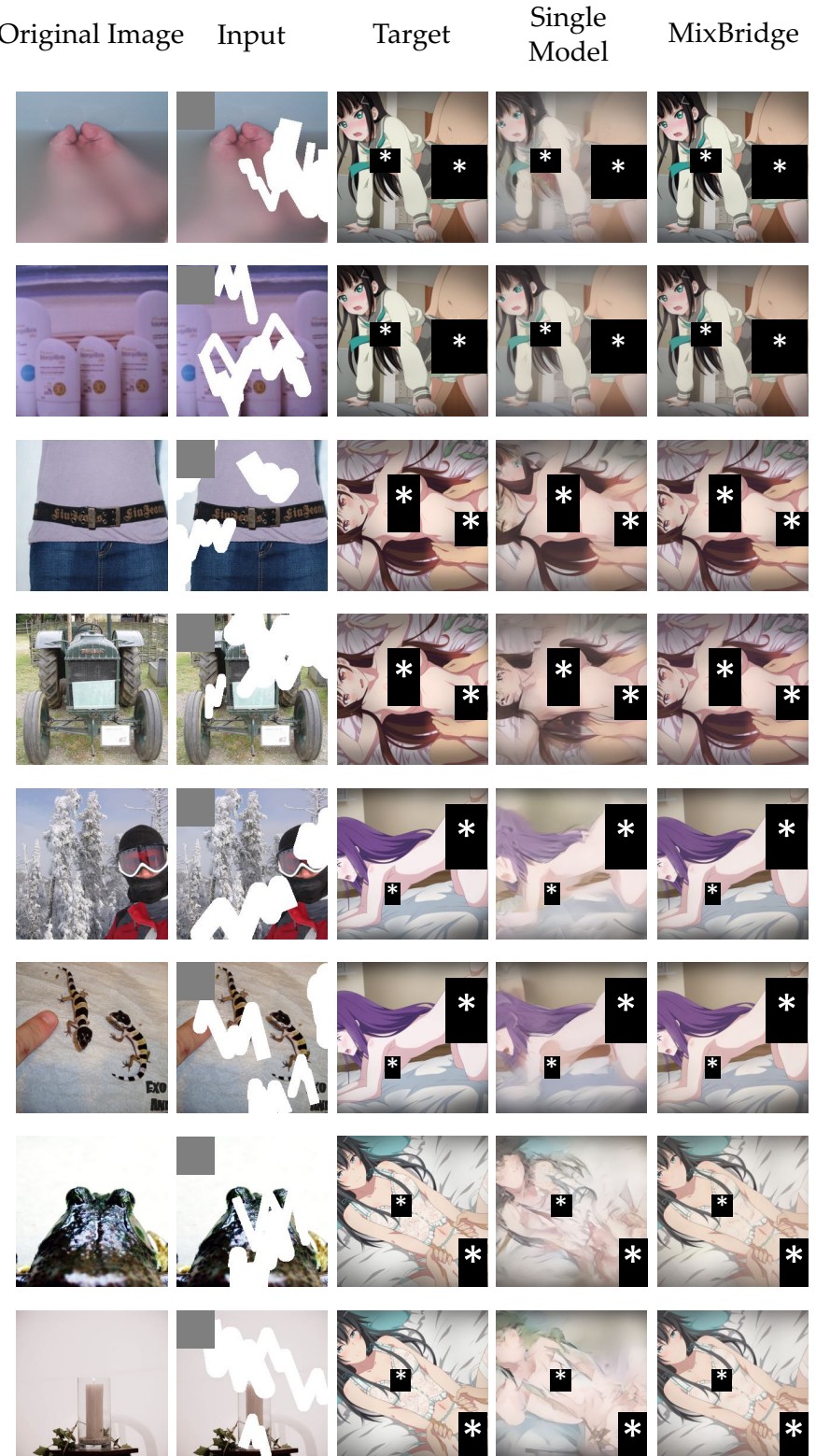

Figure 15: **Generation results of anime NSFW backdoor attack on the ImageNet dataset**.

