# OpenReview forum: "MixBridge: Heterogeneous Image-to-Image Backdoor Attack through Mixture of Schrödinger Bridges"
_ICML.cc/2025/Conference — ICML 2025 poster_

### Official Review · Reviewer_LwZQ · 2025-03-11

**Overall Recommendation:** 3

**Summary:**

The author proposed the MixBridge framework to conduct backdoor attacks on the Image-to-Image diffusion bridge model (called the Image-to-image Schrodinger bridge (I2SB)). Existing methods are homogeneous attacks, while MixBridge can achieve heterogeneous attacks. Specifically, MixBridge introduces the Mixture of Experts (MoE) mechanism into the backdoor trigger injection stage, and designs a Weight Reallocation Scheme to prevent the weight assignment from being too far from uniform, so as to achieve covert attacks. The author verified the effectiveness of MixBridge on two tasks: super-resolution and image inpainting.


## update after rebuttal
Although the author claims to have already discussed the relationship between the existing methods and the proposed backdoor attack, I didn't find it in the "2. Related Work" section. I will maintain my original rating.

**Claims And Evidence:**

The author assesses the stealthiness of backdoor attacks using Entropy, and judges whether the backdoor trigger is covert through Weight Average (as shown in Figure 2). Is this a common approach? Is there any literature support? In section 5.2. Evaluation Metrics of the paper, there is no mention of relevant literature to illustrate this point.

**Essential References Not Discussed:**

None.

**Experimental Designs Or Analyses:**

Yes.

1. It can be seen from Table 1 that compared with the benign model (I2SB), the images generated by the MixBridge attack proposed in this paper have higher quality. This advantage seems counterintuitive. Please explain the source of this performance improvement when introducing the backdoor while improving the performance of the generative model.
2. The experimental part of the paper does not compare with cutting-edge methods.

**Methods And Evaluation Criteria:**

Yes.

**Other Comments Or Suggestions:**

1. In section 2. Related Work of the paper, the relationship between existing methods and the method in this paper is not discussed.
2. In the caption of Figure 2, the double quotes of "Weight Average" need to be modified.
3. The meaning represented by the background color of the cells in Table 1 (optimal and suboptimal values) should be clearly explained.
4. Please add a discussion on the defense methods against the MixBridge backdoor attack by the author to address potential security issues.

**Other Strengths And Weaknesses:**

None.

**Questions For Authors:**

1. Add comparisons and discussions of cutting-edge methods in the experimental section.
2. Please explain why the generated image quality of the backdoored model in Table 1 is better than that of the benign model.

**Relation To Broader Scientific Literature:**

None.

**Theoretical Claims:**

Yes. The author conducted a theoretical analysis of the "limitations of using a single I2SB model for heterogeneous backdoor attacks".

---

> ### Author Rebuttal · Authors · 2025-04-01
>
> We sincerely express our gratitude for your acceptance and constructive comments. Due to space constraints, we include `additional_tables.pdf` and visualizations at the following link: https://anonymous.4open.science/r/ICML25-5787/. References to **Table x** and **Figure x** in our responses correspond to the materials provided in this link. The responses to your questions and weaknesses are listed below:
>
> > Q1 Alternative evaluation methods for stealthiness
>
> **A1:** Entropy has been widely adapted to measure privacy in the security field [1,2]. Greater entropy indicates an increasing difficulty in distinguishing between different events and thus a better "anonymity". Similarly, we use entropy to measure the anonymity of these experts. The higher the entropy, the harder it is to distinguish these experts, and thus the backdoor experts are more difficult to be detected. As another way to measure stealthiness, we evaluate the performance of each expert on benign tasks. If any expert consistently performs poorly on benign tasks, it could be easily detected and removed by the user. In particular, we input clean image $x_t^c$ to simulate the benign generation task and get the output $\epsilon_i(x_t^c,t;\theta_i)$ of each expert. Next, we predict the corresponding $x_0^{\epsilon_i}$ using Eq. 1 in our paper. Finally, we compute the MSE between $x_0^{\epsilon_i}$ and clean images. The results are as follows:
>
> |  | Entropy | Benign Expert MSE(E-02) | Backdoor Expert1 MSE(E-02) | Backdoor Expert2 MSE(E-02) | Backdoor Expert3 MSE(E-02) |
> |---|---|---|---|---|---|
> | MixBridge w/o WRS | 3e-05 | 0.46 | 31.87 | 33.85 | 46.35 |
> | MixBridge w WRS | 1.99 | 3.80 | 1.20 | 1.45 | 1.35 |
>
> The results show the backdoor experts generate noticeably different contents in the benign generation task if WRS is not applied. On the contrary, all experts in MixBridge with WRS produce outputs similar to the benign image. We have provided the visualization results in Figure 6 in the paper. In addition, the results of the **direct** metric (MSE) are consistent with those from the **indirect** metric (entropy).
>
> > Q2 MixBridge has better performance
>
> **A2:** We have noticed the MixBridge outperforms a single I2SB in the super-resolution task. The major reason is we apply a MoE mechanism to train the model, which brings much **stronger capacity**. Specifically, it helps mitigate the conflicts between the benign generation task and multiple backdoor attacks, enabling MixBridge to achieve better performance.
>
> > Q3 Compare with cutting-edge methods
>
> **A3:** To the best of our knowledge, there are **no** backdoor attack methods designed for I2I diffusion generation models. The setting of the unconditional diffusion model, which starts the diffusion process from a standard Gaussian noise, may not be applicable to the I2I framework. Here, we take two commonly used backdoor attack methods for standard unconditional diffusion models, BadDiffusion [3] and VillanDiffusion [4], to compare with our proposed MixBridge in the I2I setting. The results are as follows:
>
> |  | FID | PSNR | SSIM(E-02) | MSE(E-02) | CLIP(E-02) |
> |---|---|---|---|---|---|
> | MixBridge | **72.28** | **25.80** | **76.14** | **0.10** | **94.34** |
> | BadDiffusion | 307.18 | 11.05 | 20.70 | 257.12 | 75.76 |
> | VillanDiffusion | 310.58 | 11.14 | 20.83 | 135.60 | 65.65 |
>
> The results show that the two backdoor attack methods designed for unconditional diffusion models perform significantly worse than MixBridge in both benign generation and backdoor attack tasks. This demonstrates backdoor attacks designed for unconditional models are not easily transferable to the I2I setting.
>
> > Q4 The relationship between existing methods and the method in this paper is not discussed.
>
> **A4:** We have already discussed the relationship between existing methods and our proposed backdoor attack. Here, we present a simple summary. Existing backdoor attacks on diffusion models can be roughly categorized into two lines: attacks on the unconditional diffusion models and attacks on the T2I diffusion models. The former injects triggers into the input Gaussian noise to generate malicious outputs, while the latter corrupts text inputs with triggers to manipulate the diffusion process. In contrast, in this paper, our method targets to conduct heterogeneous backdoor attacks against I2I bridge-based diffusion models.
>
> We will revise the Related Work section to clearly articulate the distinctions between existing methods and our proposed backdoor attacks.
>
> > Q5 Possible defense
>
> **A5:** We present the discussion about defense methods in the response to the **Reviewer 8an4-Q1**.
>
> In addition, we sincerely appreciate your meticulous review! We will revise these typos and clarify the corresponding expressions.
>
> > Reference:
> >
> > [1] Quantifying and measuring anonymity
> > [2] Towards measuring anonymity
> > [3] How to backdoor diffusion models?
> > [4] Villandiffusion: A unified backdoor attack framework for diffusion models

---

> > ### Comment · Reviewer_LwZQ · 2025-04-03
> >
> > Although the author claims to have already discussed the relationship between the existing methods and the proposed backdoor attack, I didn't find it in the "2. Related Work" section. I will maintain my original rating.

---

> > > ### Author Response · Authors · 2025-04-04
> > >
> > > We apologize for the insufficient discussion about the relationship between the existing methods and the proposed backdoor attack due to space constraints. Below, we provide a comprehensive literature review.
> > >
> > > Backdoor attacks introduce hidden vulnerabilities into a model during training, enabling attackers to manipulate outputs at inference using a specific trigger. Early studies on backdoor attacks mainly focus on the **classification tasks** [1,2], where a backdoored model produces predefined predictions only when the input contains the trigger. The corresponding defense algorithms include backdoor detection [3] and trigger recovery [4]. More recently, researchers have investigated **backdoor attacks on generative models**. For example, [5] explores backdoor attacks in GANs, while [6] designs attacks for I2I networks.
> > >
> > > Regarding the **diffusion model**, previous studies alter the diffusion process by injecting triggers into the input Gaussian noise. In the "2. Related Work" section of our manuscript, we have roughly discussed the two categories of existing backdoor attacks against diffusion models, the attacks on unconditional diffusion models and T2I diffusion models. Some researchers have also proposed defenses for backdoored diffusion models. [7] computes the KL divergence between the trigger and a standard Gaussian noise to detect the backdoor. [8] inverse the trigger by minimizing the gap between the triggered generation and the original trigger.
> > >
> > > In this paper, we have proposed a novel backdoor attack for MixBridge. Our key contributions are listed as follows:
> > >
> > > - We conduct backdoor attacks on an **I2I diffusion Schrödinger bridge model**. To the best of our knowledge, this is the first study on backdoor attacks in I2I diffusion models.
> > > - We propose a new type of backdoor attack, **heterogeneous backdoor attack**. Unlike prior backdoor attacks, we implant multiple backdoors into the model, enabling diverse backdoor attacks.
> > >
> > > Based on the introductions above, the **relationship** between our proposed backdoor attack and prior studies can be concluded as follows. In terms of the **similarities**, our backdoor attack method is built upon the generative diffusion framework, which solves I2I tasks in the benign case, and generates malicious outputs if the input contains a trigger. However, it differs in four key aspects. **First**, unlike early backdoor attacks that induce misclassifications, the MixBridge aims to generate target backdoor images. **Second**, the MixBridge targets the bridge-based diffusion models that directly take images as inputs, while previous studies mainly focus on unconditional or T2I diffusion models that start from a standard Gaussian noise. **Third**, we consider heterogeneous backdoor attacks against diffusion models, while previous studies consider a single backdoor attack only. **Forth**, existing defenses for diffusion models rely on the assumption that the diffusion process begins with Gaussian noise. As a result, they may not effectively mitigate our proposed I2I backdoor attack.
> > >
> > > In conclusion, we propose a heterogeneous backdoor attack method against the I2I bridge-based diffusion model. We will incorporate this discussion into our manuscript.
> > >
> > > > Reference:
> > > >
> > > > [1] Badnets: Identifying vulnerabilities in the machine learning model supply chain
> > > > [2] Lira: Learnable, imperceptible and robust backdoor attacks
> > > > [3] Practical detection of trojan neural networks: Data-limited and data-free cases
> > > > [4] Neural cleanse: Identifying and mitigating backdoor attacks in neural networks
> > > > [5] The devil is in the GAN: backdoor attacks and defenses in deep generative models
> > > > [6] Backdoor Attacks against Image-to-Image Networks
> > > > [7] Ufid: A unified framework for input-level backdoor detection on diffusion models
> > > > [8] Elijah: Eliminating backdoors injected in diffusion models via distribution shift
> > > >

---

### Official Review · Reviewer_45tw · 2025-03-12

**Overall Recommendation:** 4

**Summary:**

This paper introduces MixBridge, a novel Diffusion Schrödinger Bridge (DSB)-based approach for enabling heterogeneous backdoor attacks on image-to-image models. The authors first demonstrate that a straightforward method—training a single DSB model with poisoned image pairs—can effectively execute such attacks. However, they identify challenges in performing heterogeneous attacks using a single DSB model. To address this, the authors propose a Mixture-of-Experts (MoE) strategy that integrates multiple DSB models, enhancing the effectiveness of heterogeneous backdoor attacks.

**Claims And Evidence:**

Yes

**Essential References Not Discussed:**

No.

**Experimental Designs Or Analyses:**

Yes. I checked the experimental sections in detail, including the normal tasks super-resolution and image inpainting, with Fake Face, NSFW, and Anime NSFW attacks.

**Methods And Evaluation Criteria:**

Yes

**Other Comments Or Suggestions:**

Nan

**Other Strengths And Weaknesses:**

The paper is well-structured, clearly presented, and easy to follow.

The studied problem—heterogeneous attacks on image-to-image (I2I) models—is important and underexplored.

The proposed approaches are insightful, well-motivated, and thoroughly evaluated. I believe this work will provide valuable insights to the research community.

**Questions For Authors:**

I have no major concerns regarding the evaluation of this paper. However, I have a minor question: Can the authors provide insights on designing effective defense strategies against the proposed attack?

**Relation To Broader Scientific Literature:**

Prior works predominantly focus on backdoor attacks on unconditional or text-to-image diffusion models. To the best of my knowledge, this work is the first to investigate backdoor attacks on image-to-image diffusion models. Moreover, heterogeneous backdoor attacks have been relatively underexplored in the literature. This study provides new insights into enabling such attacks, making a valuable contribution to the field.

**Theoretical Claims:**

I read the theoretical sections but did not examine the proofs in detail.

---

> ### Author Rebuttal · Authors · 2025-04-01
>
> We sincerely thank you for accepting our work and for your constructive comments. Due to space constraints, we include `additional_tables.pdf` and visualizations at the following link: https://anonymous.4open.science/r/ICML25-5787/. References to **Table x** and **Figure x** in our responses correspond to the materials provided in this link. The responses to your questions and weaknesses are listed below:
>
> > Q1 Possible defense
>
> **A1:** To the best of our knowledge, existing defense mechanisms are primarily focused on T2I diffusion models and unconditional diffusion models. In addition, they mainly focus on a single expert model for backdoor attacks. In contrast, our proposed attack targets the **bridge-based I2I diffusion model** with **heterogeneous** MoE backdoor attacks. Thus, previous defense mechanisms may not be applicable to our setting.
>
> Here, we conduct some experiments to investigate if previous defense mechanisms can be adapted to the I2I framework. We take Elijah [1] as an example to detect the trigger for a simple MixBridge model with two experts. Elijah assumes that the backdoor attack redirects the clean distribution $x_c^t\sim\mathcal{N}(\mu_c^t,\cdot)$ to the backdoor distribution $x_b^t\sim\mathcal{N}(\mu_b^t,\cdot)$ at step $t$ using a trigger $\tau$. The trigger can be optimized via the following formula.
>
> $\tau=\arg\min_\tau\|\mathbb{E}_{\epsilon\sim\mathcal{N}(0,1)}[M(\epsilon+\tau,T)]-\lambda\tau\|.$
>
> $M$ represents the model, and $\lambda$ is a hyperparameter related to the diffusion process.
>
> However, the original Elijah defense is built upon the assumption that the generation process starts from a standard Gaussian noise (i.e., $\mu_b^T=0$). In the I2I scenario, we propose a **modified version of Elijah**. Assume the gap between the benign and backdoor distributions caused by $\tau$ to be expressed as:
> $$\mu_b^t-\mu_c^t=\lambda^t\tau.$$
> According to the Eq. 1 in our paper, we derive the following objective.
>
> $\tau=\arg\min_\tau\|\sigma_1\mathbb{E}[\epsilon_{\text{Mix}}(x_1^c+\tau,t=1;\theta_{\text{Mix}})-\epsilon_{\text{Mix}}(x_1^c,t=1;\theta_{\text{Mix}})]-\lambda\tau\|.$
>
> Ideally, one should first generate the trigger and finetune the model with the Elijah method, and perform backdoor attacks again to evaluate if the defense is effective. We compare the attack performance of the generated trigger with that of the original baseline.
>
> | Models | MSE(E-02) | CLIP(E-02) |
> |---|---|---|
> | MixBridge | **0.10** | **94.34** |
> | Elijah | 32.70 | 60.03 |
> | Modified Elijah | 32.64 | 59.30 |
>
> It turns out that the triggers generated by the defense methods fail to manipulate the diffusion process. In other words, they cannot inverse the trigger in the MixBridge, let alone defend against the attack. We attribute such failures to the complex distribution in the I2I generation process. In this case, the image distribution gap cannot be simply modeled by a trigger $\tau$. In addition, Elijah does not solve the **heterogeneous** backdoor. Please refer to **Figure 2** for the visualizations of triggers generated by Elijah.
>
> > Reference:
> >
> > [1] Elijah: Eliminating backdoors injected in diffusion models via distribution shift

---

> > ### Comment · Reviewer_45tw · 2025-04-02
> >
> > I thank the authors for their reply. All my concerns have been addressed. I will keep my original rating.

---

> > > ### Author Response · Authors · 2025-04-05
> > >
> > > Thank you sincerely for your review and comments. We are deeply grateful for your recognition of our work's contributions and your acknowledgment that our responses addressed your concerns. The discussion about backdoor attack defense has significantly strengthened the paper, and we truly appreciate the time and expertise you dedicated to evaluating this research.

---

### Official Review · Reviewer_8an4 · 2025-03-13

**Overall Recommendation:** 3

**Summary:**

This paper introduces MixBridge, a novel diffusion Schrödinger bridge (DSB) framework designed to implant multiple heterogeneous backdoor triggers in bridge-based diffusion models, which accommodate complex and arbitrary input distributions. Unlike prior backdoor approaches that focus on single-attack scenarios with Gaussian noise inputs, MixBridge enables backdoor injection through direct training on poisoned image pairs, eliminating the need for complex modifications to stochastic differential equations. the authors propose a Divide-and-Merge strategy, where models are pre-trained for individual backdoors and later integrated into a unified model. Additionally, a Weight Reallocation Scheme (WRS) enhances the stealthiness of MixBridge. Empirical evaluations demonstrate the effectiveness of the proposed framework across diverse generative tasks.

**Claims And Evidence:**

Yes

**Essential References Not Discussed:**

No.

**Experimental Designs Or Analyses:**

Yes, the authors conducted extensive experiments.

**Methods And Evaluation Criteria:**

Yes. The two metrics used in this paper, utility and specificity, are widely used in related works.

**Other Comments Or Suggestions:**

No

**Other Strengths And Weaknesses:**

Strength:
1. This paper propses a new backdoor attack against diffusion model.
2. Conduct extensive experiments to evaluate the proposed approach.

Weakness:
1. This authors do not evaluate the performance of the proposed approach under some defense mechanism,

**Questions For Authors:**

Could you consider some SOTA defense mechanism in the evluation section?

**Relation To Broader Scientific Literature:**

The key contributions of this paper build upon and extend multiple strands of existing research in backdoor attacks, diffusion models, and Schrödinger bridge frameworks. Prior work on backdoor attacks has primarily focused on single-trigger scenarios, often in classification models, with limited exploration in diffusion-based generative frameworks. Additionally, most backdoor studies have relied on Gaussian noise as the input distribution, restricting their applicability to more complex data settings. This paper broadens the scope by introducing MixBridge, which allows for backdoor implantation in bridge-based diffusion models handling arbitrary input distributions, thereby generalizing beyond previous Gaussian-based approaches.

**Theoretical Claims:**

I checked the proof in Appendix B, and do not find any issues.

---

> ### Author Rebuttal · Authors · 2025-04-01
>
> We sincerely thank you for accepting our work and for your constructive comments. Due to space constraints, we include `additional_tables.pdf` and visualizations at the following link: https://anonymous.4open.science/r/ICML25-5787/. References to **Table x** and **Figure x** in our responses correspond to the materials provided in this link. The responses to your questions and weaknesses are listed below:
>
> > Q1 Possible defense
>
> **A1:** To the best of our knowledge, existing defense mechanisms are primarily focused on T2I diffusion models and unconditional diffusion models. In addition, they mainly focus on a single expert model for backdoor attacks. In contrast, our proposed attack targets the **bridge-based I2I diffusion model** with **heterogeneous** MoE backdoor attacks. Thus, previous defense mechanisms may not be applicable to our setting.
>
> Here, we conduct some experiments to investigate if previous defense mechanisms can be adapted to the I2I framework. We take Elijah [1] as an example to detect the trigger for a simple MixBridge model with two experts. Elijah assumes that the backdoor attack redirects the clean distribution $x_c^t\sim\mathcal{N}(\mu_c^t,\cdot)$ to the backdoor distribution $x_b^t\sim\mathcal{N}(\mu_b^t,\cdot)$ at step $t$ using a trigger $\tau$. The trigger can be optimized via the following formula.
>
> $\tau=\arg\min_\tau\|\mathbb{E}_{\epsilon\sim\mathcal{N}(0,1)}[M(\epsilon+\tau,T)]-\lambda\tau\|.$
>
> $M$ represents the model, and $\lambda$ is a hyperparameter related to the diffusion process.
>
> However, the original Elijah defense is built upon the assumption that the generation process starts from a standard Gaussian noise (i.e., $\mu_b^T=0$). In the I2I scenario, we propose a **modified version of Elijah**. Assume the gap between the benign and backdoor distributions caused by $\tau$ to be expressed as:
> $$\mu_b^t-\mu_c^t=\lambda^t\tau.$$
> According to the Eq. 1 in our paper, we derive the following objective.
>
> $\tau=\arg\min_\tau\|\sigma_1\mathbb{E}[\epsilon_{\text{Mix}}(x_1^c+\tau,t=1;\theta_{\text{Mix}})-\epsilon_{\text{Mix}}(x_1^c,t=1;\theta_{\text{Mix}})]-\lambda\tau\|.$
>
> Ideally, one should first generate the trigger and finetune the model with the Elijah method, and perform backdoor attacks again to evaluate if the defense is effective. We compare the attack performance of the generated trigger with that of the original baseline.
>
> | Models | MSE(E-02) | CLIP(E-02) |
> |---|---|---|
> | MixBridge | **0.10** | **94.34** |
> | Elijah | 32.70 | 60.03 |
> | Modified Elijah | 32.64 | 59.30 |
>
> It turns out that the triggers generated by the defense methods fail to manipulate the diffusion process. In other words, they cannot inverse the trigger in the MixBridge, let alone defend against the attack. We attribute such failures to the complex distribution in the I2I generation process. In this case, the image distribution gap cannot be simply modeled by a trigger $\tau$. In addition, Elijah does not solve the **heterogeneous** backdoor. Please refer to **Figure 2** for the visualizations of triggers generated by Elijah.
>
> > Reference:
> >
> > [1] Elijah: Eliminating backdoors injected in diffusion models via distribution shift

---

### Official Review · Reviewer_DGbD · 2025-03-14

**Overall Recommendation:** 3

**Summary:**

The paper presents a framework for injecting multiple heterogeneous backdoor triggers into bridge-based diffusion models. The authors propose a "Divide-and-Merge" strategy, where backdoors are trained independently and later integrated using an MoE framework. Additionally, a Weight Reallocation Scheme (WRS) is introduced to improve the stealthiness of backdoor attacks. Empirical results on ImageNet and CelebA demonstrate the effectiveness of MixBridge in both benign and backdoor generation scenarios.

**Claims And Evidence:**

The claim “existing backdoor formulations are designed for generative models that exclusively take Gaussian noise as input” is not valid. There are existing backdoor attacks for I2I model with super-resolution images as input, such as [1].

[1] Jiang, Wenbo, et al. "Backdoor Attacks against Image-to-Image Networks." arXiv preprint arXiv:2407.10445 (2024).

**Essential References Not Discussed:**

The paper lacks discussion on existing backdoors for the I2I model (see Claims and Evidence). In addition, it is also encouraged to discuss possible defenses (by adapting existing ones) that may or may not defend MixBridge.

**Experimental Designs Or Analyses:**

1. The experimental setup is sound. However, the paper could provide more details on hyperparameter selection (e.g., trade-off parameter in Eq. 9).

2. Some backdoor defenses for T2I diffusion models such as Elijah [1] could be adapted for I2I diffusion models. The authors are encouraged to discuss possible defenses on MixBridge.

[1] An, Shengwei, et al. "Elijah: Eliminating backdoors injected in diffusion models via distribution shift." Proceedings of the AAAI Conference on Artificial Intelligence. Vol. 38. No. 10. 2024.

**Methods And Evaluation Criteria:**

1. While the chosen metrics are standard in image generation and attack evaluation, the paper does not explore alternative evaluation methods that might capture stealthiness more effectively beyond entropy measurements.

2. Moreover, the stealthiness of the trigger added to images may need further consideration. If the trigger is too obvious, it could be easily detected, reducing the effectiveness of the backdoor attack in real-world scenarios.

3. It’s not clear why a two-stage approach is necessary. Could the Weight Reallocation Scheme loss L_WRS be incorporated during the router's training in the first stage?

**Other Comments Or Suggestions:**

None.

**Other Strengths And Weaknesses:**

The paper in general is well-written with clear motivation. The method and evaluation are properly designed.

In addition to the lack of discussion on literature, I have one additional concern regarding **Attacker’s Capacity and Goal**. If the attacker is the model owner (with full control of the training process), I am not clear under which practical scenarios, the model owner would want to realize such a backdoor to make downstream users produce specific NSFW/low-quality images. As a model owner, I can understand the need to use backdoor techniques to realize some useful functions like watermarking or personalization. But here, why would a model provider want to make its users generate corrupted outputs which clearly brings it no benefit. The authors should provide more discussion on the practical significance of the proposed attack.

**Questions For Authors:**

None.

**Relation To Broader Scientific Literature:**

This paper contributes to the growing literature on backdoor attacks/defenses in I2I diffusion models.

**Theoretical Claims:**

Proposition 4.1 argues that the natural pairing of images in I2SB facilitates backdoor training without explicit SDE modifications. Theorem 4.2 rigorously proves that a single I2SB model approximates the geometric mean of benign and backdoor distributions, leading to performance conflicts. They are aligned with their method and experiment results.

---

> ### Author Rebuttal · Authors · 2025-04-01
>
> We sincerely express our gratitude for your acceptance and constructive comments. Due to space constraints, we include `additional_tables.pdf` and visualizations at the following link: https://anonymous.4open.science/r/ICML25-5787/. References to **Table x** and **Figure x** in our responses correspond to the materials provided in this link. Below are our responses to your questions and identified weaknesses:
>
> > Q1 Existing backdoor attacks for I2I model
>
> **A1:** We acknowledge some previous studies have explored backdoor attacks on I2I models. However, **none** of them focus on diffusion-based models, let alone those based on the Schrödinger Bridge. We will restrict our discussion in the introduction to the diffusion-based models, and clarify the key differences from the previously mentioned I2I backdoor methods.
>
> > Q2 Alternative evaluation methods for stealthiness
>
> **A2:** We have included the answer in our response to **Reviewer LwZQ-Q1**.
>
> > Q3 The stealthiness of the trigger needs further consideration
>
> **A3:** Here, we evaluate the stealthiness based on trigger size. A **smaller** trigger is generally **harder** to detect. In particular, we conduct super-resolution task along with three heterogeneous backdoor attacks on the CelebA dataset using different trigger sizes in **Table 1**.
>
> According to the experiments, the ASR remains 0.93 for a $16\times16$ trigger size (i.e., $1\%$ of the total image), which is a small trigger with relatively high stealthiness. In contrast, BadDiffusion [1] uses a $14\times14$ trigger on a $32\times32$ image, which occupies nearly $20\%$ of the image area. We present visualizations of different trigger sizes in **Figure 1** in the anonymous link.
>
> > Q4 Why not one-stage WRS
>
> **A4:** Following previous studies [2,3], we pre-train each expert separately and then finetune them jointly. Here, we conduct an ablation study for the one-stage method.
>
> |  |  | FID | PSNR | SSIM(E-02) | MSE(E-02) | CLIP(E-02) | ASR | Entropy |
> |:---:|:---:|:---:|:---:|:---:|:---:|:---:|:---:|:---:|
> | one-stage | w/o WRS | 148.29 | 24.74 | 72.36 | 1.78 | 73.23 | 68.21 | 4e-04 |
> |  | WRS | 111.30 | 24.04 | 74.33 | 1.75 | 79.65 | 83.46 | **2.00** |
> | two-stage (Ours) | w/o WRS | **74.67** | 25.01 | **74.64** | 0.96 | **93.68** | 98.53 | 3e-05 |
> |  | WRS | 92.00 | **25.43** | 74.27 | **0.64** | 93.21 | **98.73** | 1.99 |
>
> Intuitively, the one-stage model significantly **underperforms** compared to the two-stage version. A possible reason is in one-stage training the experts might suffer from **conflicting** optimization directions across different generation tasks in the early phase, leading to poor and suboptimal solutions. By using a two-stage method, the pre-training process naturally avoids such issues, and makes the finetuning process more stable and effective.
>
> > Q5 More details on hyperparameter
>
> **A5:** As a trade-off hyperparameter, $\lambda$ controls the intensity of WRS. In this paper, to ensure the stealthiness of the backdoor attacks, we chose a large $\lambda$ to obtain high entropy. Here, we investigate how $\lambda$ affects the performance of the MixBridge.
>
> | $\lambda$ | FID | PSNR | SSIM(E-02) | MSE(E-02) | CLIP(E-02) | ASR | Entropy |
> |:---:|:---:|:---:|:---:|:---:|:---:|:---:|:---:|
> | 1 | 92.00  | 25.43  | 74.27  | **0.64**  | **93.21**  | **98.73**  | **1.99**  |
> | 0.1 | 90.47  | 25.53  | 73.73  | 0.75  | 92.61  | 98.38  | 1.93  |
> | 0.07 | 87.77  | 24.90  | 73.74  | 0.70  | 92.14  | 97.98  | 1.65  |
> | 0.05 | 76.38  | 25.29  | 75.01  | 0.67  | 92.86  | 98.13  | 1.24  |
> | 0.03 | 75.20  | **25.85**  | **75.98**  | 0.90  | 92.38  | 97.73  | 0.13  |
> | 0.01 | **70.45**  | 25.02  | 74.92  | 1.07  | 91.24  | 98.19  | 0.01  |
>
> The results show that a lower $\lambda$ improves benign generation but degrades the performance of backdoor attacks. We find that setting $\lambda=0.1$ or 1 leads to a relatively good tradeoff.
>
> > Q6 Possible Defenses
>
> **A6:** Please refer to our response to **Reviewer 8an4-Q1**.
>
> > Q7 Attacker’s Capacity and Goal
>
> **A7:**  We apologize for the unclear explanation of the backdoor attack settings. On one hand, the model owner (or the attacker) may not be primarily motivated by financial gain. Instead, their intentions could include deliberate sabotage, reputation damage, or other non-economic incentives. For example, an attacker might aim to create chaos, undermine competitors’ credibility, or provoke legal and ethical issues to gain a strategic advantage. On the other hand, as you mentioned, backdoor techniques can be used for legitimate purposes like watermarking or personalization. Therefore, this paper can also potentially promote the studies for legitimate applications.
>
> We will revise our corresponding claim in the paper according to your suggestion.
>
> > Reference:
> > [1] How to backdoor diffusion models?
> > [2] Warm: On the benefits of weight averaged reward models
> > [3] Mogu: A framework for enhancing safety of open-sourced llms while preserving their usability

---

### Decision · Program_Chairs · 2025-05-01

**Decision:**

Accept (poster)

**Comment:**

The paper presents MixBridge, a novel framework for executing heterogeneous backdoor attacks in bridge-based diffusion models, specifically within the Diffusion Schrödinger Bridge (DSB) framework. The proposal introduces two core innovations: the Divide-and-Merge strategy—where backdoors are independently trained and merged using a Mixture-of-Experts (MoE) architecture—and a Weight Reallocation Scheme (WRS) that enhances the stealthiness of the attack by promoting balanced expert contributions.

Across all four reviews, the paper is commended for addressing an underexplored area in backdoor attack research—heterogeneous backdoor injection—especially in the context of image-to-image generative tasks. The reviewers also appreciate the simplification the paper offers over traditional approaches, by directly training on poisoned image pairs rather than modifying underlying stochastic differential equations. The proposed MoE strategy is seen as a practical and modular solution for combining multiple backdoors.

Overall, I believe this paper presents solid contribution to the community and I would recommend Accept.